# `ConStellaration`: A dataset of QI-like stellarator plasma boundaries and optimization benchmarks

**Santiago A. Cadena**    **Andrea Merlo**    **Emanuel Laude**    **Alexander Bauer**
**Atul Agrawal**    **Maria Pascu**    **Marija Savtchouk**    **Enrico Guiraud**
**Lukas Bonauer**    **Stuart Hudson**    **Markus Kaiser**

Proxima Fusion
{scadena, amerlo}@proximafusion.com

## Abstract

Stellarators are magnetic confinement devices under active development to deliver steady-state carbon-free fusion energy. Their design involves a high-dimensional, constrained optimization problem that requires expensive physics simulations and significant domain expertise. Recent advances in plasma physics and open-source tools have made stellarator optimization more accessible. However, broader community progress is currently bottlenecked by the lack of standardized optimization problems with strong baselines and datasets that enable data-driven approaches, particularly for quasi-isodynamic (QI) stellarator configurations, considered as a promising path to commercial fusion due to their inherent resilience to current-driven disruptions. Here, we release an open dataset of diverse QI-like stellarator plasma boundary shapes, paired with their ideal magnetohydrodynamic (MHD) equilibria and performance metrics. We generated this dataset by sampling a variety of QI fields and optimizing corresponding stellarator plasma boundaries. We introduce three optimization benchmarks of increasing complexity: (1) a single-objective geometric optimization problem, (2) a "simple-to-build" QI stellarator, and (3) a multi-objective ideal-MHD stable QI stellarator that investigates trade-offs between compactness and coil simplicity. For every benchmark, we provide reference code, evaluation scripts, and strong baselines based on classical optimization techniques. Finally, we show how learned models trained on our dataset can efficiently generate novel, feasible configurations without querying expensive physics oracles. By openly releasing the dataset (`https://huggingface.co/datasets/proxima-fusion/constellaration`) along with benchmark problems and baselines (`https://github.com/proximafusion/constellaration`), we aim to lower the entry barrier for optimization and machine learning researchers to engage in stellarator design and to accelerate cross-disciplinary progress toward bringing fusion energy to the grid.

## 1   Introduction

Fusion energy promises virtually limitless, carbon-free power by harnessing the same process that powers the sun. Magnetic confinement fusion approaches trap a fully ionized gas (plasma) within magnetic fields to sustain the conditions required for fusion. Among these, *stellarators* confine the plasma solely through external coils, which produce three-dimensional, twisted magnetic flux surfaces (Figure 1). Unlike *tokamaks*, stellarators do not rely on large internal plasma currents, thereby avoiding associated instabilities [1, 2]. However, this advantage comes with a

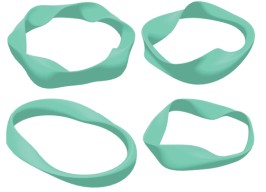

Figure 1: Examples of diverse stellarator plasma boundaries.

39th Conference on Neural Information Processing Systems (NeurIPS 2025) Track on Datasets and Benchmarks.

trade-off. Designing stellarators involves a significantly more complex parameter space: shaping the three-dimensional plasma boundary to satisfy multiple physics and engineering constraints is a high-dimensional, constrained, optimization problem.

Stellarator design has been mainly approached as a two-stage process [3]. In *stage one*, the magnetic field that confines the plasma is optimized; in *stage two*, electromagnetic coils are designed to reproduce this field. *Stage one*, the focus of this work, optimizes a three-dimensional surface that defines the boundary condition for the plasma equilibrium magnetic field. The surface is commonly parameterized by a truncated Fourier series in cylindrical coordinates (Figure 2, left). A solution to the ideal-magnetohydrodynamics (MHD) equations is then computed to determine the magnetic field inside the plasma [4]. VMEC [5] and its recent C++ re-implementation [6] are classical physics codes that compute a solution to the ideal-MHD model (Figure 2). From the MHD solution, we can compute multiple magnetic field properties, e.g. the *rotational transform*, that we can iteratively optimize to target a desired value in an outer optimization loop by updating the plasma boundary.

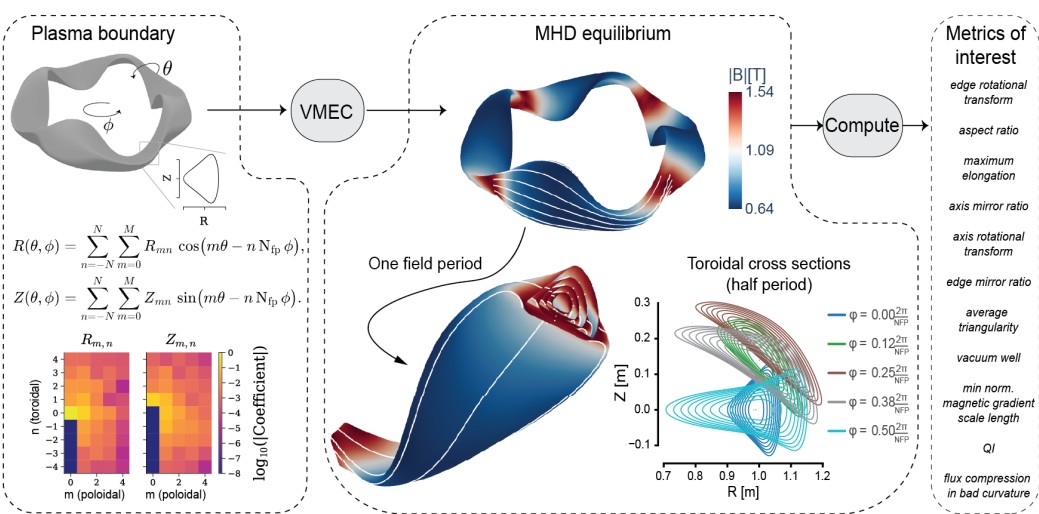

Figure 2: A plasma boundary is defined by the coefficients $R_{mn}$ and $Z_{mn}$ of a truncated Fourier series in cylindrical coordinates, parametrized by the lab-frame poloidal angle $\theta$ and toroidal angle $\phi$. This boundary is passed to the VMEC++ code [5, 6] to compute an ideal-MHD equilibrium. In this example, the configuration is stellarator symmetric, meaning that $R(\theta, \phi) = R(-\theta, -\phi)$ and $Z(\theta, \phi) = -Z(-\theta, -\phi)$, and the number of repeated field periods ($N_{\mathrm{fp}}$) is four. The ideal-MHD equilibrium defines the magnetic field throughout the plasma volume, comprising nested magnetic flux surfaces on which magnetic field lines (depicted in white) lie. We can then compute various metrics of interest from the equilibrium field.

Unlike tokamaks, classical or unoptimized stellarators lack toroidal symmetry and inherently suffer from poor confinement of high-energy particles: such fusion-born particles often escape the plasma volume, striking plasma facing components before depositing their energy back into the plasma. This prevents a self-sustained fusion process. The root cause lies in the behavior of particles trapped in poloidal, toroidal, or helical magnetic wells, which fail to sample the entire magnetic flux surface, experiencing a net radial drift that leads to gradually loosing confinement. These challenges are a direct consequence of the non-axisymmetric magnetic geometry of stellarators. A particularly effective strategy to suppress these drifts is to optimize stellarators imposing the condition of *omnigeneity* [7, 8], which requires only that the average radial drift of trapped particles vanishes. Among omnigenous fields, quasi-isodynamicity (QI) fields have poloidally closed contours of the magnetic field strength [9–11], which results in a vanishing net plasma toroidal current. The advantages of even approximate QI fields have been validated in laboratory experiments, most notably in the Wendelstein-7X (W7-X) stellarator [12]. These compelling benefits have made the QI symmetry a target in the design of next-generation stellarator-based fusion power plants [13, 14].

Major advances in open-source software frameworks for stellarator design have been presented in recent years. For example, SIMSOPT [15] provides high-level interfaces to link plasma equilibrium solvers such as VMEC [5] or SPEC [16] with numerical optimizers. Moreover, tools like DESC [17]

have leveraged end-to-end automatic differentiation [18] to simultaneously compute MHD equilibria and target desired properties. However, these tools still present a high entry barrier for practitioners in the optimization and machine learning communities, as they require substantial domain knowledge to make meaningful contributions.

Although significant progress has been made in defining what to target in stellarator design, there remains a lack of standardized benchmark problems and evaluation protocols to address stage one optimization. This contrasts to other areas of machine learning, where well-defined challenges have driven rapid and measurable progress [19]. Establishing such benchmarks in stellarator research would offer significant value by enabling systematic comparisons of optimization methods across a range of problem formulations. For instance, different representations (parameterizations) of the plasma boundary may vary in their effectiveness: some may better avoid local minima, while others may facilitate faster or more reliable convergence to feasible solutions. Our contributions are as follows.

- We release a diverse dataset of about 158,000 QI-like stellarator plasma boundaries with their associated ideal-MHD equilibria (in vacuum and for five different levels of plasma beta, the ratio between the plasma thermal pressure and the magnetic pressure) computed with `VMEC++` [6] and corresponding figures of merit.
- We propose three optimization problems of varying complexity and kind, and release associated code.
- We provide a set of baselines for these optimization problems using classical optimization approaches.
- We show that models trained on our dataset can generate novel configurations that satisfy optimization constraints, even when only a handful of training examples do.

## 1.1   Related work

**Stellarator datasets.**   Beyond works releasing a small set of plasma configurations [20, 21], large-scale datasets have focused on stellarators with quasi-axisymmetry (QA) or quasi-helical symmetry (QH) [22–24], but not QI. These studies rely on an expansion about the magnetic axis [25–30] (the field line representing the innermost flux surface) that reduces the 3D MHD equations into a 1D ordinary differential equation, which is much faster to solve. Landreman [22] sampled $\sim 500k$ QA and QH configurations, while Giuliani [23], Giuliani et al. [24] sampled $\sim 370k$ QA and QH configurations as part of the `QUASR` dataset [1]. To the best of our knowledge, none of these datasets include publicly available computed ideal-MHD equilibria.

**Stellarator optimization benchmarks.**   While several studies have proposed sets of optimization problems to test optimization strategies or shape parameterizations (e.g.[31]), and others have surveyed optimization approaches [32], there are no standardized benchmarks for stellarator optimization.

## 1.2   Background

In *Boozer* coordinates [33] (Figure 3), magnetic field strength contours of QI fields exhibit three characteristic properties: (i) the contours close poloidally, appearing as vertically closed loops in a Boozer plot; (ii) the magnetic field strength maxima align along straight vertical lines; and (iii) the arc length between points of equal magnetic field strength along a field line depends only on the flux surface (i.e., it is invariant across field lines) [11]. The targets in Figure 4 and Figure 5 are examples of precise QI fields.

## 2   A diverse dataset of QI-like plasma boundaries and ideal-MHD equilibria

Directly sampling the Fourier coefficients representing the plasma boundary (Figure 2) would very rarely lead to good (or even valid) stellarator fields [34]. To generate a large and diverse dataset of stellarator configurations that are approximately QI, we aim to sample diverse QI fields and other geometrical properties, and search for plasma boundaries that produce those target fields. These target

---

[1]`https://quasr.flatironinstitute.org`

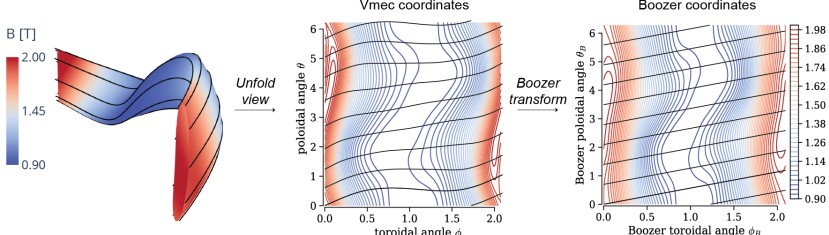

Figure 3: Visualization of the iso-contours of the magnetic field strength $B$ and a few magnetic field lines (black). In the *Boozer* coordinate system [33], the original poloidal and toroidal angles are transformed into Boozer angles $\theta_B$ and $\phi_B$, respectively, to straighten the magnetic field lines (black).

generative factors include the aspect ratio $A$ (the ratio between the major and minor toroidal radii: $R_0/a$), the edge rotational transform $\iota_{edge}$ (how far a field line moves around the "short" (poloidal) way along the torus each time it goes once around the "long" (toroidal) way), the mirror ratio $\Delta_{edge}$ (defined as $(B_{\max} - B_{\min})/(B_{\max} + B_{\min})$), and the maximum elongation $\epsilon_{\max}$ (the largest cross-section elongation across toroidal angles [11]).

For a given target QI field and set of properties, we generated surfaces either through physics-informed heuristics (Section 3 of Goodman et al. [11]), fast near-axis expansion models [29, 35] (using pyQSC [2]), or through *stage-one* optimization runs. We passed all resulting surfaces to our forward model running VMEC++ at high fidelity (Section 3) to obtain ideal-MHD equilibria and metrics of interest (Figure 2). All configurations are limited to poloidal and toroidal mode numbers of at most four. Assuming stellarator symmetry, $R_{m,n} = 0; m = 0, n < 0$ and $Z_{m,n} = 0; m = 0, n \le 0$, and fixing the major radius $R_{0,0} = 1$, the total number of degrees of freedom is 80 (Figure 2, left).

**Sampling targets.** To sample diverse QI fields, we used the parameterization for an omnigenous field from Dudt et al. [8], imposing stellarator symmetry (Section A.1). Notably, our fields span a diverse range of magnetic well shapes and show variation in how these wells are stretched along field lines (Figure 4). The other target properties were drawn from a uniform distribution spanning a range of sensible values (Table 6).

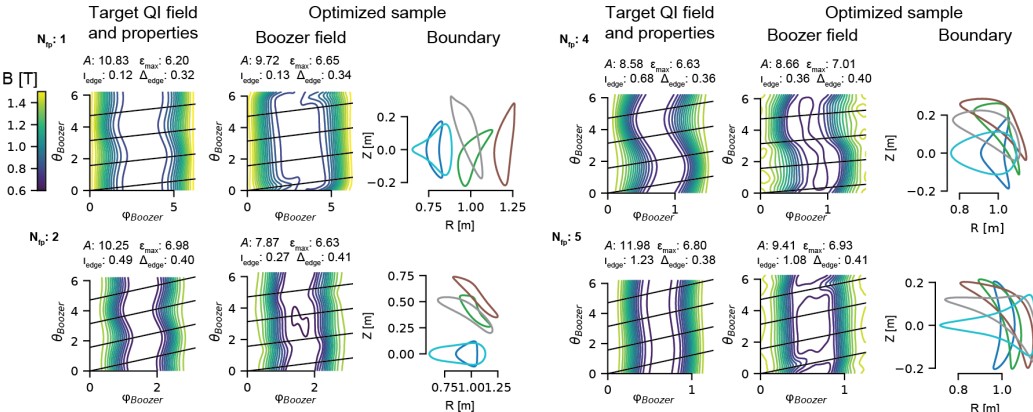

Figure 4: Four optimized samples from our dataset with 1, 2, 4, and 5 field periods. A finite computational budget for each sample generation leads to an approximate QI field at the plasma boundary. All field plots share the same color bar and the boundary cross-section labels correspond to those in Figure 2.

**Optimization.** We implemented stage-one optimization approaches seeded with heuristic or near-axis expansion models using DESC or VMEC++ -based frameworks and varying objective settings (Section A.2). Multiple optimization approaches with finite budget increased diversity in the resulting

---
[2]https://github.com/rogeriojorge/pyQIC

boundaries, even for the same set of target field and properties (Figure 5). Each DESC run took three minutes on average on a 32 vCPU 128GB RAM machine, while each VMEC++ run took around one hour on average on a 32 vCPU 32GB RAM machine.

**Results.** We began by sampling 100k target sets. From this pool, we generated 30k and 49k plasma boundary candidates using our heuristic and near-axis-expansion models, respectively. We then applied the DESC optimizer twice to each target–once per initialization strategy–yielding an additional 88k optimized boundaries. A subset of 15k targets was also optimized with VMEC++ in the loop, seeded by rotating ellipses. Altogether, this produced roughly 182k candidate configurations, and we evaluated equilibria and metrics with the high-fidelity forward model on 158k of them without errors. Among these successful cases, 15k, 20k, 68k, 27k, and 28k configurations have 1, 2, 3, 4, and 5 field periods, respectively. Our resulting dataset spans a broad range of target metrics (Fig. 6, left) and reveals strong correlations between prescribed targets and the achieved values (Figure 6, right). To enable investigations of equilibrium properties at finite pressure profiles (i.e., beyond vacuum), we also made available the ideal-MHD equilibria of the boundaries at five different volume-averaged $\beta$ values [3] $(1, 2, 3, 4,$ and $5\%)$ and their correspondent metrics.

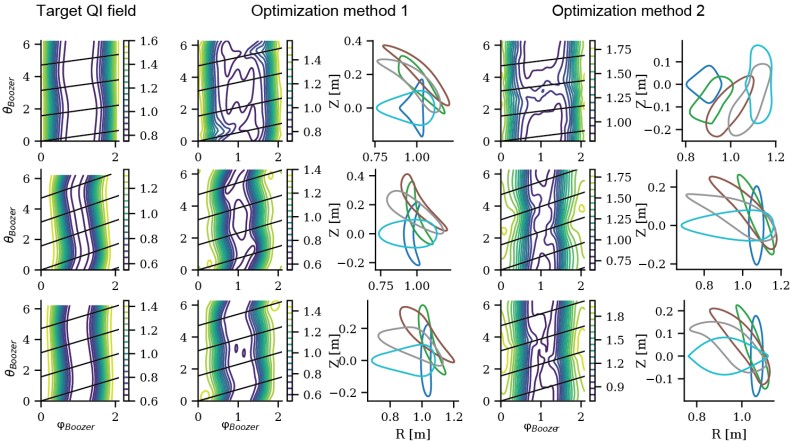

Figure 5: Diverse plasma configurations obtained for the same targets. Optimization methods vary in initialization strategy, framework, and settings. While some runs favor matching the target QI field and mirror ratio, other runs better match the remaining target properties.

**Statistical consistency of the dataset.** To assess the degree to which the target metrics can be inferred from the available boundary coefficients, we trained an ensemble of multilayer perceptrons (MLP) model on the optimized dataset samples and evaluated its performance on a held-out test set. The model achieved good predictive accuracy ($R^2 > 0.97$ and RMSE<0.1 for all metrics; Section A.4), indicating that the metrics are fairly learnable in-domain with an expressive enough model.

## 3    Optimization benchmark

Stellarator design can be naturally formulated as a multi-objective constrained optimization problem [32]. The objectives and constraints arise from both engineering and economic considerations (e.g., limiting the aspect ratio to achieve a compact device) as well as physics-based requirements (e.g., ensuring a stable MHD plasma). The design process involves translating stakeholder expectations into a consistent set of feasible requirements, and navigating the trade-offs among conflicting objectives in a manner that aligns with the overarching design goals.

We introduce three prototypical stellarator design tasks with increasing complexity each involving different subsets of design metrics (Table 1): (1) *Geometric*, (2) *Simple-to-Build QI*, and (3) *MHD-stable QI*, detailed in Table 2.

---

[3] $\beta$ is defined as the ratio of the thermal plasma pressure $p$ to the magnetic field pressure that has to be externally applied: $\beta = 2\mu_0 p/B^2$ where $\mu_0$ is the vacuum permeability. To set these beta values, we assumed a radial linear pressure profile and scaled the pressure at the axis to match the target volume-averaged beta for each boundary.

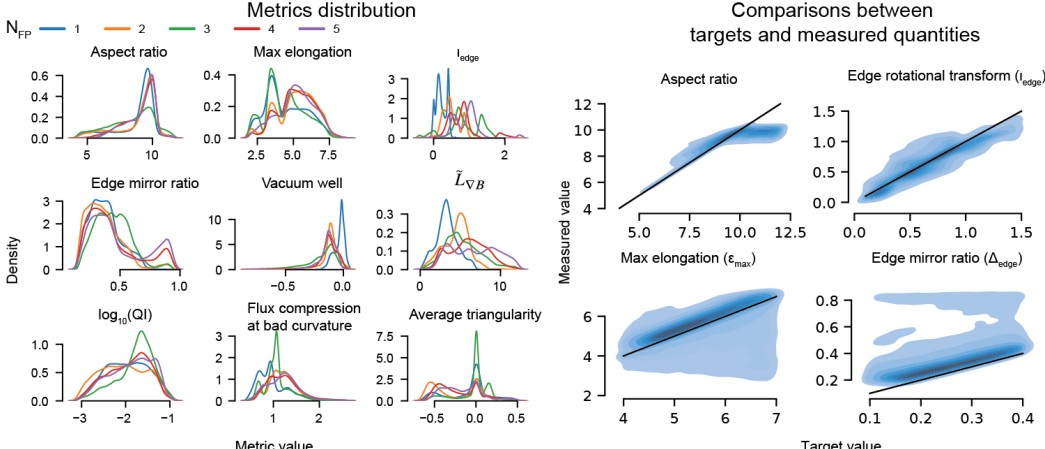

Figure 6: Distribution of metrics and comparisons between targets and outcomes. Pair plots show only optimized configurations. Black lines represent the identity. $A$, $\epsilon_{max}$, and $\Delta_{edge}$ were used as upper-bound constraints during optimization, while the rotational transform was enforced as equality constraint.

**Forward Model**   We leverage VMEC++ [6] to compute vacuum 3D ideal-MHD equilibria, scaled to $R_0 = 1\,\mathrm{m}$, $B_0 \simeq 1\,\mathrm{T}$. Each vacuum equilibrium is fully defined by a single flux-surface mapping $\Sigma_\Theta : (\theta, \varphi) \longmapsto (R, \phi, Z)$, where $\theta$ and $\varphi$ are generic poloidal and toroidal angles, respectively, and $\Theta$ denotes the set of surface parameters, and $(R, \phi, Z)$ are cylindrical coordinates. In VMEC++ , the truncated Fourier series presented in Figure 2 is used for $\Sigma_\Theta$. However, for the purpose of these optimization benchmark problems, we make no assumptions about the functional form of $\Sigma_\Theta$. All optimization problems have the form:

$$
\begin{aligned}
\min_{\Theta} \quad & (f_1(\Theta), f_2(\Theta), \dots) \\
\text{subject to} \quad & c_i(\Theta) \leq c_i^*, \ \forall i \,,
\end{aligned}
\tag{1}
$$

where $f_i : \mathbb{R}^D \to \mathbb{R}$ are objective functions, $c_i : \mathbb{R}^D \to \mathbb{R}$ are constraint functions, and $c_i^*$ are constraint targets. Each objective and constraint depends directly on the magnetic field, which in turn is determined by the surface mapping that defines the boundary condition of the ideal-MHD model.

| (a) **Geometric problem** | (b) **Simple-to-build QI** | (c) **MHD-stable QI** |
|---|---|---|
| $\min_{\Theta} \ \epsilon_{\max}$ | $\min_{\Theta} \ -\widetilde{L}_{\nabla B}$ | $\min_{\Theta} \ \left(-\widetilde{L}_{\nabla B}, \ A\right)$ |
| s.t. $\ A \leq A^*,$ | s.t. $\ \tilde{\iota} \geq \tilde{\iota}^*, \ QI \leq QI^*$ | s.t. $\ \tilde{\iota} \geq \tilde{\iota}^*, \ QI \leq QI^*$ |
| $\bar{\delta} \leq \bar{\delta}^*,$ | $\Delta \leq \Delta^*, \ A \leq A^*$ | $\Delta \leq \Delta^*, \ W_{\mathrm{MHD}} \geq 0$ |
| $\tilde{\iota} \geq \tilde{\iota}^*.$ | $\epsilon_{\max} \leq \epsilon_{\max}^*$ | $\langle \chi_{\nabla r} \rangle \leq \langle \chi_{\nabla r} \rangle^*$ |

Table 2: Constrained optimization problem formulations. See Table 1 for semantic associations to the symbols. All metrics are a function of the boundary

## 3.1   Problem 1: Geometric

To onboard contributors to stellarator optimization, we propose an intuitive, purely geometric problem (Table 2) where we look for stellarators that minimize the maximum elongation $\epsilon_{max}$ for a given aspect ratio $A$, edge rotational transform $\tilde{\iota}$, and average triangularity $\bar{\delta}$. $\bar{\delta}$ averages the triangularity between the two

| Metric | Acronym |
|---|---|
| minimum normalized magnetic gradient scale length | $\widetilde{L}_{\nabla \mathbf{B}}$ |
| edge rotational transform over number of field periods | $\tilde{\iota}$ |
| aspect ratio | $A$ |
| max elongation | $\epsilon_{\max}$ |
| edge magnetic mirror ratio | $\Delta$ |
| quasi isodynamicity residual | $QI$ |
| vacuum well | $W_{MHD}$ |
| flux compression in regions of bad curvature | $\langle \chi_{\nabla r} \rangle$ |
| average triangularity | $\bar{\delta}$ |

Table 1: Equilibrium field metrics and their acronyms.

stellarator-symmetric cross-sections ($\phi = 0$ and $\phi = \pi/N_{\text{fp}}$), and $\tilde{\iota}$ is the edge rotational transform per field period.

## 3.2 Problem 2: Single-objective simple-to-build QI stellarator

Stellarators are notoriously challenging to construct due to their inherently three-dimensional magnetic geometry. Optimized designs like W7-X demand millimeter coil tolerances [36]. Moreover, the development and assembly of such devices can run into cost and schedule overruns driven by manufacturing complexity, potentially leading to the cancellation of entire projects as it was the case for the NCSX stellarator [37, 38]. This raises a key question: *Can optimized QI stellarators be realized using simpler, easier-to-manufacture coils?*

In a fusion reactor, the spatial region between the plasma and coils must accommodate a divertor, first wall (plasma-facing material components), neutron shielding, tritium-breeding blanket, and magnets structural support. These layers, together with the magnet superconducting technologies (e.g., low-temperature superconductors (LTS) or high-temperature superconductors (HTS)), impose geometric and engineering constraints on coil design. The feasibility of a stellarator configuration depends not only on plasma performance but also on how easily the required magnetic fields can be generated using manufacturable coils.

Not all magnetic fields are equally *coil-friendly*. We colloquially refer to *coil simplicity* as the ease with which modular coils can be placed and shaped to produce the desired field. For example, surfaces with high coil simplicity allow coils to be located further from the plasma and require lower curvature and fewer tight bends. We quantify coil simplicity using the normalized magnetic field gradient scale length on the plasma boundary, following Kappel et al. [39]. This metric has proven effective in guiding optimization towards configurations with simpler, more feasible coil designs [13, 14].

Historically, QI stellarators have required particularly complex coil geometries compared to other quasi-symmetric configurations [40–43]. This benchmark problem challenges that assumption by optimizing for *precise* QI fields that can be generated with *simple* coils.

Table 2 introduces the problem definition, where $\widetilde{L}_{\nabla \mathbf{B}}$ is magnetic field gradient scale length [39] normalized by $a/N_{\text{fp}}$, $QI = \frac{1}{4\pi^2} \int \int r_{QI}^2 \, \mathrm{d}\theta \, \mathrm{d}\phi$ quantifies deviation from a *precise* QI field following Goodman et al. [11], and $\Delta$ is the magnetic mirror ratio at the plasma boundary. We normalize the objective by $a/N_{\text{fp}}$ to ensure scale invariance across configurations with varying field period numbers. Since a QI field is easier to achieve for large aspect ratio configurations, highly elongated flux surfaces, and large mirror ratios [11], we explicitly control these quantities through inequality constraints.

## 3.3 Problem 3: Multi-objective ideal-MHD stable QI stellarators

This optimization problem introduces two new critical constraints for reactor relevant stellarator design: ideal-MHD plasma stability and mitigation of turbulent transport.

Despite the fact that QI configurations eliminate current-driven instabilities ("disruptions") that often affect tokamak designs, pressure-driven instabilities persist [9], thus limiting access to high fusion power density regimes. To optimize for ideal-MHD stability, we adopt the vacuum magnetic well $W_{\text{MHD}}$ as a proxy [25, 44]

Turbulent transport, expected to be dominated by ion-temperature gradient (ITG) turbulence in QI stellarators [12, 45, 46], limits the achievable fusion gain. Landreman et al. [47] demonstrated how purely geometrical quantities correlate strongly with the turbulence heat flux. As a constraint, we compute the "flux-surface compression in regions of *bad curvature*" given by $\chi_{\nabla r} = \mathcal{H}(\mathbf{B} \times \kappa \cdot \nabla \alpha) \|\nabla r\|_2^2$ as a simple geometric proxy. Here $\mathcal{H}$ is the Heavyside step function, $\mathbf{B} \times \kappa \cdot \nabla \alpha$ is the curvature drift [48], $\kappa$ is the magnetic field curvature, $\alpha$ is the field line label, and $\nabla r$ is the flux compression. A positive curvature drift represents regions of *bad curvature*. This quantity is evaluated on a single-flux surface at $\rho = r/a = 0.7$.

In quasi-poloidal (QP) and QI stellarators, $L_{\nabla \mathbf{B}} \propto R_0/N_{\text{fp}} = aA/N_{\text{fp}}$ [4]. More compact devices (low $A$) reduce capital cost per unit output power [49, 50] but increase coil complexity (proxied by $L_{\nabla \mathbf{B}}$). This

---

[4]Assuming that the characteristic length scale of the magnetic field gradient satisfies $L_{\nabla \mathbf{B}} \propto L_{\nabla B}$, and considering a QP magnetic field where the magnetic field strength forms a single well, i.e., $B(\varphi) = B_0 \cos(N_{\text{fp}}\varphi)$, where $\varphi$ is a field-aligned coordinate.

trade-off motivates a Pareto-optimal search [51] between coil simplicity and compactness. Table 2 introduces the problem definition, where $\langle \cdot \rangle$ denotes flux-surface averaging.

### 3.4 Evaluation metric

We release evaluation code that scores candidate plasma boundaries across benchmarks. Our evaluation code requires the plasma boundaries to be represented by the truncated Fourier series in cylindrical coordinates (see Figure 2).

**Single-objective scoring** For single-objective problems, we map each design point to a bounded scalar score value $s(\Theta)$ given by:

$$s(\Theta) = \begin{cases} h\big(f(\Theta)\big) & \text{if } \tilde{c}_i(\Theta) \leq \varepsilon, \ \forall i, \\ 0 & \text{otherwise,} \end{cases} \tag{2}$$

where $f(\Theta)$ is the objective value, $h : \mathbb{R} \to [0, 1]$ is a linear map that rescales objectives into the $[0, 1]$ interval (higher is better), $\tilde{c}_i = (c_i - c_i^*)/c_i^*$ is the $i$-th normalized constraint violation, and $\varepsilon$ is a relative tolerance.

**Multi-objective scoring** For multi-objective problems, we compute the hypervolume (HV) indicator [52, 53] over feasible solutions (i.e., those with $\tilde{c}_i(\Theta) \leq \varepsilon, \ \forall i$ ) using a fixed reference point in objectives space.

## 4 Optimization baselines

We now provide baselines for the three optimization problems. For all experiments, we target stellarators with three field periods and seed optimizations from rotating ellipse configurations. For the single-objective case (problem 1 and 2), we benchmark three approaches: a) gradient-based (where the gradient of the objective and constraint functions is approximated via forward finite-differences) trust-region interior point constrained optimizer [54] (`scipy-trust-constr`); b) gradient-free `COBYQA` [55] algorithm (`scipy-COBYQA`); and c) Augmented Lagrangian method (ALM) [56, 57] with a non-Euclidean proximal regularization [58–60] employing the `NGOpt` gradient-free meta-algorithm from Nevergrad [61] (`ALM-NGOpt`), to solve the subproblem. Implementation specifics are provided in the Section A.5.

Only `ALM-NGOpt` obtains feasible solutions, while both `scipy-trust-constr` and `scipy-COBYQA` did not (Table 4 and 3). Consequently, our leaderboard (Table 3) reports results exclusively for `ALM-NGOpt`. Figure 9 shows the optimized QI field and a representative coilset for the simple-to-build problem.

| Problem | Score ↑ |
|---|---|
| Geometrical | 0.969 |
| Simple-to-build | 0.431 |
| MHD-stable | 130.0 |

Table 3: `ALM-NGOpt` scores.

The multi-objective problem is decomposed into multiple single-objective problems by treating the aspect ratio as an inequality constraint. Using `ALM-NGOpt`, we found solutions for four of these instances. A sparse Pareto front is provided in Figure 7.

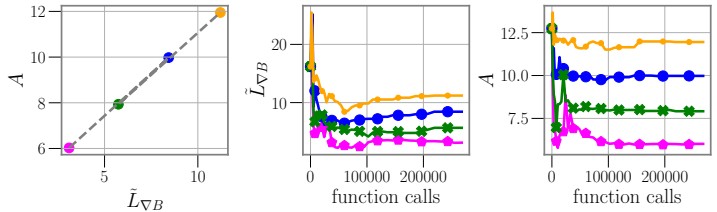

Figure 7: Pareto-front for the multi-objective optimization problem of MHD stable QI stellarators obtained with `ALM-NGOpt`.

## 5 Generative modeling of feasible domains without access to the oracle

We present a method to generate feasible configurations using learning-based models trained on the dataset, without relying on a zero-order oracle (e.g., `VMEC++` ) and with limited feasible examples.

| Method | Simple-to-build | | Geometric problem | |
|---|---|---|---|---|
| | $\tilde{L}_{\nabla B}$ ↑ | norm. constr. viol. | $\epsilon_{max}$ ↓ | norm. constr. viol. |
| `scipy-trust-constr` | 2.10* | 3.25* | 15.0* | 0.301* |
| `scipy-COBYQA` | 14.4* | 2.04* | 1.27 | 0.953 |
| `ALM-NGOpt` | 8.61 | 0.009 | 1.27 | 0.0002 |

Table 4: Comparison of baselines for the simple-to-build and the geometric problem. ↑ means that a quantity is maximized and ↓ means that a quantity is minimized. Final optimized boundaries for which `VMEC++` failed to converge at high fidelity (i.e., the fidelity with which we score a plamsa boundary) are represented with *; for them, we report the objective and constraint values from a lower fidelity equilibrium computation. `scipy-trust-constr` and `scipy-COBYQA` do not produce feasible solutions. SciPy-based optimizers ran for ~ 40 hours on a machine with 4 vCPUs. `ALM-NGOpt` ran on a 96 vCPU machine for 18 hours (geometric problem) and 34 hours (simple-to-build).

| $A$ ↓ | $\tilde{L}_{\nabla B}$ ↑ | norm. constr. viol. |
|---|---|---|
| 6.02 | 2.98 | 0.104 |
| 7.93 | 5.60 | 0.00130 |
| 9.98 | 8.45 | 0.0 |
| 11.9 | 11.1 | 0.00210 |

Table 5: Objectives and constraint violations for `ALM-NGOpt` on the multi-objective problem. Optimization was carried out by solving a sequence of single-objective problems, converting one objective into a constraint $A \leq A^*$ with $A^* \in \{6, 8, 10, 12\}$. All instances were run on a 96-vCPU machine for 15–24 h.

We test whether this method can produce many valid configurations to support downstream tasks like optimization.

We reduce the input dimensionality using Principal Component Analysis (PCA) [62] to obtain a low-dimensional latent space. In this space, Random Forest classifiers [63, 64] estimate the probability that a configuration is feasible. Thresholding this probability (e.g., above 0.8) defines a soft feasible region. Within this region, we fit a Gaussian mixture model (GMM) to capture the distribution of feasible points. Treating the GMM as a prior and the classifier output as a quasi-likelihood, we use adaptive Markov chain Monte Carlo (MCMC) [9, 65] to sample from the posterior. This allows us to generate several new configurations that are likely to satisfy constraints without querying the oracle ( Figure 8). Details are given in Section A.6 with full algorithmic details in Algorithm 2.

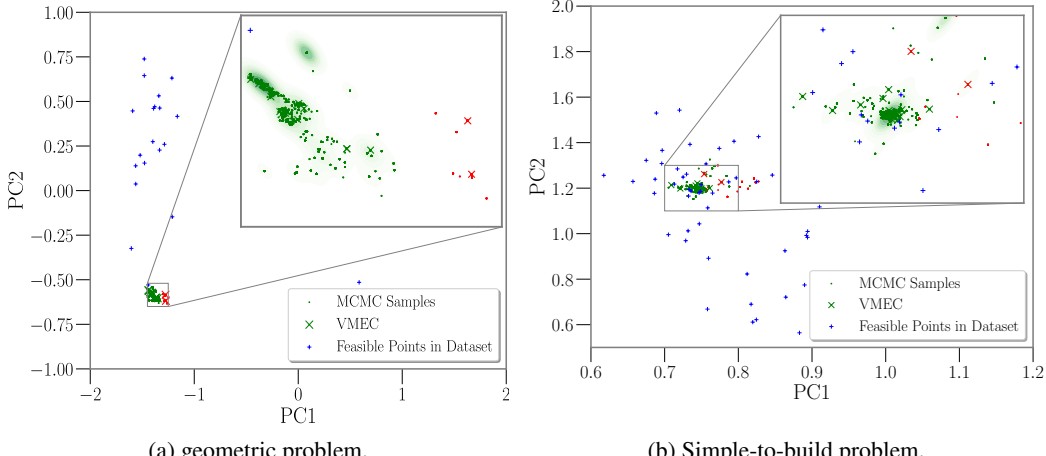

(a) geometric problem.  (b) Simple-to-build problem.

Figure 8: Posterior estimate of the feasible region in the first two PCA dimensions for two constraint-relaxed problems. Blue crosses represent feasible configurations from the dataset. Green dots show MCMC samples predicted to be feasible (classifier confidence ≥ 0.99), and red dots indicate predicted infeasible samples. Green contours reflect the estimated density of feasible samples. Oracle validation of randomly selected MCMC points are marked with green (feasible) and red (infeasible) crosses. Both the geometric and simple-to-build problems are initially relaxed, with 41 and 52 feasible points available in the dataset (out of ~ 160k).

When applied to relaxed versions of both the Geometric and Simple-to-build problems, our method successfully identifies regions of design space in which sample points are judged feasible by both the Random Forest classifier and the oracle model (i.e., using `VMEC++` ) (Figure 8).

## 6 Discussion

We released a diverse dataset of approximately $158k$ QI-like stellarator plasma boundaries, associated metrics, and ideal-MHD equilibria in vacuum and at five levels of plasma beta values. Alongside the dataset, we introduced a set of stellarator optimization tasks with strong classical baselines, designed to facilitate rigorous and reproducible evaluation of stellarator optimization strategies. We further

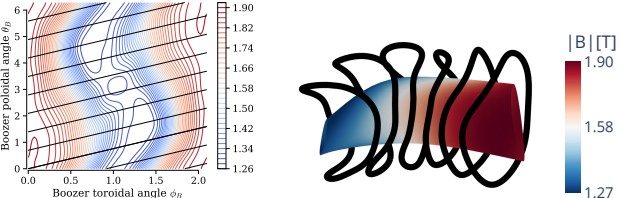

Figure 9: Left: Optimized QI magnetic field contours at the plasma boundary in Boozer coordinates for the simple-to-build optimization problem. Right: A representative coilset designed to reproduce the target magnetic field.

demonstrated a data-driven generative approach that can produce feasible plasma configurations without querying an expensive physics oracle. Nonetheless, several limitations remain. First, the degree of QI in the dataset is inherently limited by the finite-budget, optimization-based sampling process used during generation. Second, the dataset is limited to plasma boundaries; while these are usually the seeds in stellarator design, a consistent design also requires many additional systems (e.g., electromagnetic coils).

## Acknowledgments

This work was independently funded by Proxima Fusion, and supported by the BMBF grant FUSKI (FKZ: 13F1012A).

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

# A Technical Appendices

## A.1 Data generation: sampling omnigenous poloidal fields

We leverage the parameterization of an *omnigenous poloidal* field from Dudt et al. [8] in which the 1D magnetic well on each flux surface is represented by a spline on the interval $[-\pi/2, \pi/2]$. The well is symmetric about its minimum, so it can simply be parameterized between $B_{min}$ and $B_{max}$. The full omnigenous field is then built by "morphing" this one-dimensional well across magnetic field lines via a computational coordinate $h$, which is expanded in a Chebyshev basis (radial index $l$) and Fourier bases (poloidal index $m$, toroidal index $n$) with coefficients $x_{lmn}$ [8]. In practice, we generate new omnigenous-poloidal fields by sampling both the spline knots and $x_{lmn}$ coefficients. To enforce stellarator symmetry (invariance under simultaneous flips of the poloidal and toroidal Boozer angles), we set the coefficients of the odd terms of the Fourier basis along the toroidal direction to zero, namely $x_{lmn} = 0 \; \forall n >= 0$.

To produce a variety of monotonically increasing 1D well shapes, we draw knot positions from Beta$(\alpha, \beta)$ cumulative distribution functions and then rescaled them to lie between $B_{min}$ and $B_{max}$. Finally, we fix the mean magnetic field at 1 T and sample the mirror ratio $\Delta$ to determine the pair $(B_{min}, B_{max})$.

The ranges from which we sample these parameters can be found in Table 6.

| Parameter | Min | Max |
|---|---|---|
| $N_{fp}$ | 1 | 5 |
| $\tilde{\iota}$ | 0.1 | 0.3 |
| $A$ | 4.0 | 12.0 |
| $\epsilon_{max}$ | 4.0 | 7.0 |
| $\alpha_{Beta}$ | 2.0 | 6.0 |
| $\beta_{Beta}$ | 2.0 | 6.0 |
| $\Delta_{edge}$ | 0.1 | 0.4 |

Table 6: Ranges of sampling parameters with both minimum and maximum values.

## A.2 Data generation: stage one optimizations using DESC [17]

Given a set of target quantities:

$$T = \left( \iota^*, \; A^*, \; E^*, \; O^* \right)$$

where

- $\iota^*$ is the desired edge rotational transform,
- $A^*$ is the target aspect ratio,
- $E^*$ is the maximum elongation,
- $O^*$ is the target omnigenous field,

we ran numerical optimizations to find a toroidal boundary surface $\Sigma$ (parameterized in a Fourier-$RZ$ basis) that simultaneously matches these goals. Note that the mirror ratio $\Delta$ is defined within $O^*$.

### A.2.1 Initial Guess Generation

An initial boundary $\Sigma_0$ is generated either by

1. **Heuristic QP model (Section III from Goodman et al. [11])**: prescribing average major radius $R_0$, aspect ratio $A^*$, elongation $E^*$, mirror ratio, torsion, and field periods; or
2. **Near-Axis Expansion (NAE) using pyQSC** [5]: specifying $A^*$, $E^*$, $\iota^*$, mirror ratio, field periods, and mode cutoffs.

This yields a smooth $\Sigma_0$ expressed in the FourierRZToroidalSurface format of DESC.

---

[5] https://github.com/rogeriojorge/pyQIC

### A.2.2 Equilibrium Solve

Starting from $\Sigma_0$, we form the DESC equilibrium object and solve the force balance

$$\mathcal{E}(\Sigma) = \text{Equilibrium}(\Psi, \Sigma, M, N)$$

and solve the magnetostatic force-balance equations using

$$\mathcal{E} \xrightarrow{\text{solve(force)}} \mathcal{E}^{\text{sol}}.$$

### A.2.3 Objective Function

On the solved equilibrium $\mathcal{E}^{\text{sol}}$, we define individual objective terms:

$$J_A(\Sigma) = \frac{R_0(\Sigma)}{a(\Sigma)}, \qquad\qquad f_A = w_A \left(J_A - A^*\right)^2, \tag{3}$$

$$J_E(\Sigma) = \max_{\varphi} \frac{b(\varphi; \Sigma)}{a(\varphi; \Sigma)}, \qquad\qquad f_E = w_E \left(J_E - E^*\right)^2, \tag{4}$$

$$J_\iota(\Sigma) = \iota\left[\mathcal{E}^{\text{sol}}\right], \qquad\qquad f_\iota = w_\iota \left(J_\iota - \iota^*\right)^2, \tag{5}$$

$$\mathbf{J}_O(\Sigma) = O\left[\mathcal{E}^{\text{sol}}, O^*\right], \qquad\qquad f_O = w_O \left\|\mathbf{J}_O\right\|_2^2, \tag{6}$$

where $a, b$ are the minor/major half-axes of the cross-section, $\varphi$ is the toroidal angle, and the omnigenous residual $\mathbf{J}_O$ is computed by the DESC `Omnigenity` objective using the target field $O^*$.

The omnigenity contribution $\left\|\mathbf{J}_O\right\|_2^2$ is given by

$$\left\|\mathbf{J}_O\right\|_2^2 = \sum_{i=1}^{N_\eta} \sum_{j=1}^{N_\alpha} w(\eta_i) \left[B_{\text{eq}}(\rho_0, \eta_i, \alpha_j) - B^*(\rho_0, \eta_i, \alpha_j)\right]^2,$$

with the poloidal weight

$$w(\eta) = \frac{\eta_{\text{weight}} + 1}{2} + \frac{\eta_{\text{weight}} - 1}{2} \cos\eta \quad \left(\text{so } w \equiv 1 \text{ if } \eta_{\text{weight}} = 1\right),$$

where $B_{\text{eq}}$ is the field strength of the equilibrium in $(\rho, \eta, \alpha)$ coordinates and $B^*$ is the perfectly-omnigenous target field generated by `OmnigenousField` as in [8]. The residuals $r_{ij} = \sqrt{w(\eta_i)}\left(B_{\text{eq}} - B^*\right)_{ij}$ are evaluated on the same $(\eta, \alpha)$-grid used by the target field.

On a solved equilibrium $\mathcal{E}^{\text{sol}}$ at a fixed flux surface $\rho = \rho_0$, we assemble a least-squares objective

$$\mathcal{L}(\Sigma) = f_A + f_E + f_\iota + f_O.$$

Internally, DESC invokes `JAX` to compute residuals, leveraging automatic differentiation to compute gradients.

The objective is then wrapped in an augmented-Lagrangian least-squares optimizer (`lsq-auglag`) [32] to minimize $\|r\|_2^2$ alongside the other terms.

### A.2.4 Constraints

To enforce vacuum equilibrium and fix global invariants, the following constraints are imposed:

$$
\begin{aligned}
R_{0,0}(\Sigma) &= 1, & &(\text{FixBoundaryR}) \\
j_\|(\Sigma) &= 0, & &(\text{CurrentDensity}) \\
p(\Sigma) &= 0, & &(\text{FixPressure}) \\
J_{\text{tor}}(\Sigma) &= 0, & &(\text{FixCurrent}) \\
\Psi(\Sigma) &= \text{const.}, & &(\text{FixPsi})
\end{aligned}
$$

where each is implemented via the corresponding DESC linear-objective wrapper.

### A.2.5   Nonlinear Optimization

We employ DESC's `lsq-auglag` optimizer [32] to solve

$$\min_{\Sigma} \ \mathcal{L}(\Sigma) \quad \text{s.t. all linear constraints,}$$

using automatic differentiation and a trust-region least-squares augmented-Lagrangian scheme. Iterations continue until convergence (up to 200 iterations by default), yielding the optimized boundary $\Sigma^*$.

Our exact implementation is available at https://github.com/proximafusion/constellaration.

### A.3   Stage one optimizations using VMEC++ [6] in the loop

We carried out optimizations using the `NGOpt` algorithm from the `Nevergrad` [6] library. To improve convergence, we preconditioned the problem using a diagonal scaling matrix as detailed in Section A.2.1. We parameterized the boundary with up to four poloidal and toroidal Fourier modes and ran the optimization on a single machine equipped with 32 vCPUs and 32GB of RAM. Each run is allocated a time budget of approximately 1 h.

The optimization minimizes the following objective function:

$$
\begin{aligned}
f(\Theta) = & \int_0^{2\pi} \int_0^{\pi/N_{\text{fp}}} (B(\theta, \phi) - B^*(\theta, \phi))^2 \, d\theta \, d\phi \\
& + \int_0^{2\pi} \left( \max_{\phi} B(\theta, \phi) - B(\theta, \phi = 0) \right)^2 \, d\theta \\
& + \left( \frac{A - A^*}{A^*} \right)^2 \\
& + \left( \frac{\iota_{\text{edge}} - \iota_{\text{edge}}^*}{\iota_{\text{edge}}^*} \right)^2 \\
& + \left( \max \left( 0, \frac{\epsilon_{\text{max}} - \epsilon_{\text{max}}^*}{\epsilon_{\text{max}}^*} \right) \right)^2 .
\end{aligned}
\tag{7}
$$

where $B$ denotes the magnetic field strength from the ideal-MHD equilibrium in Boozer coordinates, and $B^*$ represents the target omnigenous magnetic field strength. The quantities $A$, $\iota_{\text{edge}}$, and $\epsilon_{\text{max}}$ correspond to the aspect ratio, edge rotational transform, and maximum elongation, respectively, with asterisks denoting their target values. The additional target on the maxima of the magnetic field strength guides the optimizer towards more QI fields.

In the optimization loop, we used VMEC++ within the forward model. To speed up the generation of the optimized boundary, we run VMEC++ at a lower resolution than the one used to score plasma boundaries in optimization benchmarks (e.g. reduced number of flux surfaces, higher required force tolerance to converge).

Due to the constrained time budget, the optimization may not fully minimize the objective function but added the desired diversity to the dataset.

Our exact implementation is available at https://github.com/proximafusion/constellaration.

### A.4   In-domain predictability of the dataset.

We filtered the vacuum data for three field period configurations that were also the result of either the DESC or VMEC optimizations. Then we filtered outliers (0.05% tails) for each of the metrics, resulting in $\sim 23k$ data points. Finally, we split the data into training and test (20%) sets.

---

[6]https://github.com/facebookresearch/nevergrad

To facilitate training, we also Z-scored the output metrics, keeping track of the training set statistics for later inference.

Using Bayesian optimization to sweep over hyperparameters like network depth, width, type of activation, and learning rate; we converged to an ensemble model of ten multi-layer perceptrons (MLPs) with three layers, 256 hidden units, and `tanh` activations. The MLPs mapped Fourier-boundary coefficients to target key metrics by minimizing mean squared error.

We obtained fairly good in-domain generalization results in terms of root mean squared error (RMSE), Pearson's correlation coefficient ($R^2$), normalized root mean squared error (NRMSE), and signal-to-noise ratio (SNR) (Table 7).

Table 7: Test-set performances of an MLP ensemble model trained to predict target metrics from boundary coefficients

| Metric | RMSE | $R^2$ | NRMSE | SNR |
|---|---|---|---|---|
| aspect_ratio | 0.090 | 0.997 | 0.050 | 808.54 |
| aspect_ratio_over_edge_rotational_transform | 0.581 | 0.993 | 0.080 | 507.73 |
| max_elongation | 0.161 | 0.994 | 0.072 | 731.78 |
| axis_rotational_transform_over_n_field_periods | 0.006 | 0.994 | 0.071 | 556.74 |
| edge_rotational_transform_over_n_field_periods | 0.006 | 0.997 | 0.052 | 985.74 |
| axis_magnetic_mirror_ratio | 0.010 | 0.974 | 0.159 | 79.72 |
| edge_magnetic_mirror_ratio | 0.013 | 0.989 | 0.102 | 244.02 |
| average_triangularity | 0.018 | 0.995 | 0.065 | 806.58 |
| vacuum_well | 0.006 | 0.998 | 0.062 | 1934.06 |
| minimum_normalized_magnetic_gradient_scale_length | 0.330 | 0.990 | 0.101 | 342.57 |
| flux_compression_in_regions_of_bad_curvature | 0.034 | 0.990 | 0.059 | 432.36 |
| log_10_qi | 0.051 | 0.982 | 0.134 | 132.22 |

We highlight that such surrogate models are prone to extrapolation errors, particularly when queried far from the data distribution they were trained on [66]. Uncertainty calibration, active learning, and physics informed strategies (among others) could be considered moving forward for effective surrogate-based optimizations [67, 68].

## A.5 Optimization baselines

### A.5.1 Implementation details and hyperparameters

In this section we provide implementation details for the optimization baseline. For the SciPy-based optimizers, we use default parameters, and set the maximum number of iterations to a large value.

We implement a variant of the proximal ALM [58] where the quadratic proximal term is replaced by a trust-region constraint. This can be seen as an instance of the anisotropic proximal ALM [59]. The modification is essential for improving convergence when using evolutionary algorithms (such as NGOpt), as it restricts the sampling of new candidate solutions to a region around the current iterate [60].

As the degrees of freedom $\Theta$ operate on different scale, we precondition the problem with a diagonal matrix $\text{diag}(\Lambda)$ where the entries $\Lambda$ decay exponentially. We define the rescaled variables as $\widetilde{\Theta} := \text{diag}(\Lambda)^{-1}\Theta$ and $\tilde{f}(\widetilde{\Theta}) := f(\text{diag}(\Lambda)\widetilde{\Theta})$ and $\tilde{c}_i(\widetilde{\Theta}) := c_i(\text{diag}(\Lambda)\widetilde{\Theta})$. In addition, we apply a base-10 logarithmic transformation to the QI constraint.

In each iteration, the algorithm alternates between primal and dual updates. For each constraint $\tilde{c}_i$, it tracks a penalty parameter $\rho_i^k$ and a Lagrange multiplier $y_i^k$. The complete algorithm is given in Algorithm 1.

---

**Algorithm 1** non-Euclidean proximal augmented Lagrangian method

---

**Require:** $\Theta^0 \in \mathbb{R}^D, \rho^0 \in \mathbb{R}^m_{++}, y^0 \in \mathbb{R}^m_+, \delta_0 > 0, 0 < \tau, \gamma < 1, \sigma > 1$ and $\delta_{\min}, \rho_{\max} > 0$

1: **for** $k \in \{0, 1, \ldots, N\}$ **do**
2:      Primal update

$$\widetilde{\Theta}^{k+1} = \underset{\widetilde{\Theta} \in B(\widetilde{\Theta}^k, \delta_k)}{\arg\min} \; \tilde{f}(\widetilde{\Theta}) + \tfrac{1}{2} \sum_{i=1}^{m} \tfrac{1}{\rho_i^k} \left( \max\{0, y_i^k + \rho_i^k \tilde{c}_i(\widetilde{\Theta}^k)\}^2 - (y_i^k)^2 \right) \qquad (8)$$

3:      dual update

$$y_i^{k+1} = \max\{0, y_i^k + \rho_i^k \tilde{c}_i(\widetilde{\Theta}^{k+1})\}$$

4:      update penalty parameters

$$\rho_i^{k+1} = \begin{cases} \rho_i^k & \text{if } \tilde{c}_i(\widetilde{\Theta}^{k+1}) \leq \tau \tilde{c}_i(\widetilde{\Theta}^k) \\ \min\{\rho_{\max}, \sigma \rho_i^k\} & \text{otherwise.} \end{cases}$$

5:      decrease trust-region

$$\delta_{k+1} = \max\{\delta_{\min}, \gamma \delta_k\}$$

6: **end for**

---

For the geometric problem we choose $\rho_i^0 = 10, \rho_{\max} = 1e9, \delta_0 = 0.5, \gamma = 0.9, \delta_{\min} = 0.05, \tau = 0.8, \sigma = 5$. The subproblem (8) is solved with NGOpt with a budget of $\min\{20.000, 1500 + k \cdot 260\}$ forward-model calls.

For the simple-to-build problem we choose $\rho_i^0 = 10, \rho_{\max} = 1e9, \delta_0 = 0.5, \gamma = 0.95, \delta_{\min} = 0.05, \tau = 0.8, \sigma = 5$. The subproblem (8) is solved with NGOpt with a budget of $\min\{20.000, 1500 + k \cdot 260\}$ forward-model calls.

For the MHD-stable problems we choose $\rho_i^0 = 10, \rho_{\max} = 1e8, \delta_0 = 0.33, \gamma = 0.95, \delta_{\min} = 0.05, \tau = 0.8, \sigma = 5$. The subproblem (8) is solved with NGOpt with a budget of $\min\{20.000, 1500 + k \cdot 300\}$ forward-model calls.

For all problems, $\Theta^0$ is a rotating ellipse configuration. We optimize up to four poloidal and toroidal Fourier modes, which results in $D = 80$ degrees of freedom. During the optimization, we run `VMEC++` at low fidelity.

### A.5.2   Additional experimental results

We provide convergence plots for the three problems obtained with `ALM-NGOpt`. Green curves represent metrics that are constrained. Red colored metrics are maximized and blue colored metrics are minimized. Gray curves correspond to metrics that are not part of the optimization problem. The blue dashed lines indicate lower bounds and the red dashed lines indicate upper bounds. Figure 11 provides plots corresponding to the single-objective problem, while Figure 12 provides a plot for one instance ($A \leq 8$) of the sequence of single-objective problems corresponding to the multi-objective problem.

In Figure 10 we show the initial and final plasma configurations for the different problems.

### A.6   Generative modeling details

We use the Random Forest classifier and the GMM implementations from `Scikit-learn` [69]. We use the Random walk Metropolis-Hastings algorithm [70] with adaptive proposal distribution [71] as the MCMC sampler. To monitor the convergence of the MCMC sampler, Figure 13 presents the log-probability of the posterior distribution evaluated at each sampled point. The rising and stabilizing log-probability indicates convergence to high-density regions. Algorithm 2 summarizes the formulation discussed in Section 5.

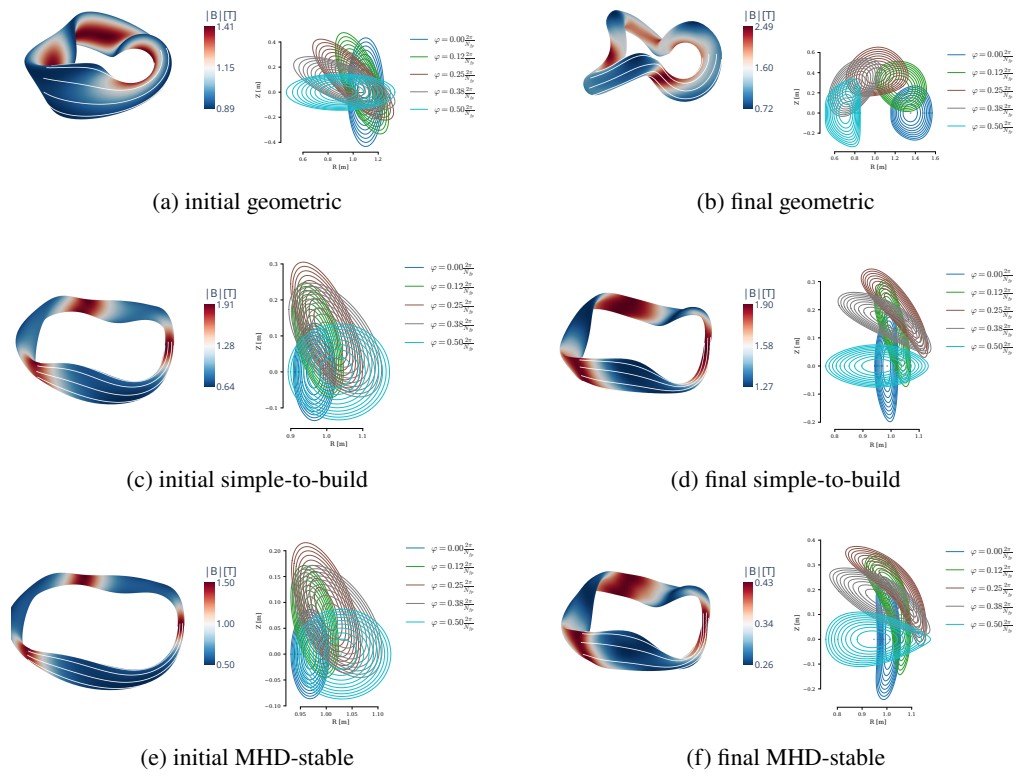

(a) initial geometric

(b) final geometric

(c) initial simple-to-build

(d) final simple-to-build

(e) initial MHD-stable

(f) final MHD-stable

Figure 10: Initial guesses and final plasma configurations optimized with ALM-NGOpt. We selected a low aspect ratio configuration from the Pareto Front of solutions for the multi-objective problem.

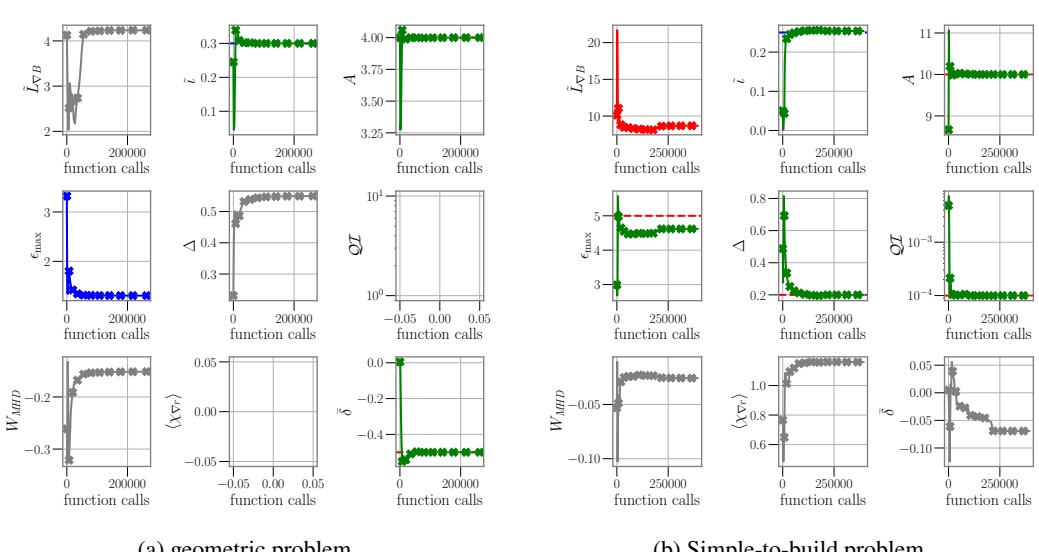

(a) geometric problem.

(b) Simple-to-build problem.

Figure 11: Single-objective problem optimization traces.

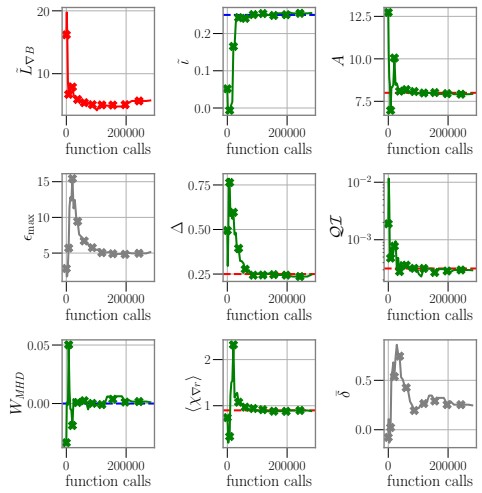

Figure 12: Multi-objective problem with $A \leq 8$.

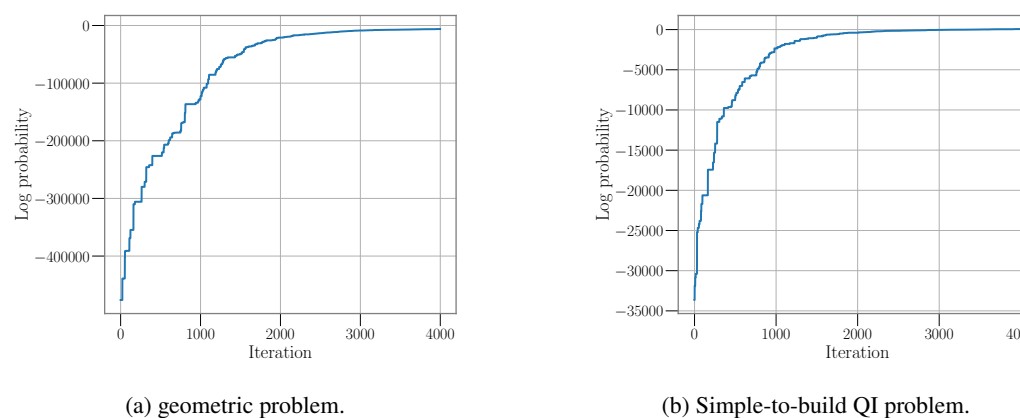

(a) geometric problem.

(b) Simple-to-build QI problem.

Figure 13: Trace plot of the log-posterior probability values over MCMC iterations.

---

**Algorithm 2** Generative Inference of Feasible Configurations without Oracle Access

---

**Require:** Dataset $\mathcal{D} = \{x_1, \ldots, x_N\} \subset \mathbb{R}^D$, constraint definition
**Ensure:** Set of configurations $\{x^*\}$ predicted to lie in the feasible domain
    // Dimensionality Reduction:
1: Compute PCA mapping $\Phi : \mathbb{R}^D \to \mathbb{R}^d$, where $d \ll D$
2: Project dataset to latent space: $Z \leftarrow \{\mathbf{z}_i = \Phi(x_i)\}_{i=1}^{N}$
    // Feasibility Classification:
3: Train Random Forest classifiers $\{C_i(\mathbf{z})\}_{i=1}^{N_c}$ to predict feasibility label $y \in \{0, 1\}$
4: Define soft-feasible region: $\tilde{\mathcal{F}} \leftarrow \bigcap_{i=1}^{N_c} \{p(C_i(\mathbf{z}) = 1) \geq \tau\}$, where $\tau_i = 0.8 \ \forall i$
    // Density Estimation:
5: Fit Gaussian Mixture Model GMM($\mathbf{z}$) on data restricted to $\tilde{\mathcal{F}}$
    // Bayesian Refinement:
6: Define prior: $p(\mathbf{z}) \leftarrow$ GMM($\mathbf{z}$)
7: Define quasi-likelihood: $\ell(\mathbf{z}) \leftarrow \sum_{i=1}^{N_c} \log C_i(\mathbf{z})$
8: Compute posterior using MCMC: $\{\mathbf{z}^*\} \sim p(\mathbf{z} \mid \text{feasible}) \propto \ell(\mathbf{z}) \cdot p(\mathbf{z})$
9: Inverse transform to original space: $x^* \leftarrow \Phi^{-1}(\mathbf{z}^*)$
    // Oracle Validation:
10: Evaluate $x^*$ using VMEC++ oracle to confirm feasibility
11: **return** $\{x^*\}$

---

