# OpenReview forum: "ConStellaration: A dataset of QI-like stellarator plasma boundaries and optimization benchmarks"
_NeurIPS.cc/2025/Datasets_and_Benchmarks_Track — NeurIPS 2025 Datasets and Benchmarks Track poster_

### Official Review · Reviewer_aH2b · 2025-06-20

**Rating:** 5
**Confidence:** 4

**Summary:**

This paper introduces a large-scale dataset and benchmark suite designed to lower the barrier for optimization and machine learning researchers to contribute to stellarator design. The authors release a dataset of approximately 158,000 QI-like stellarator plasma boundaries, each paired with its computed ideal-MHD equilibrium and performance metrics. To standardize the evaluation of new methods, they propose three optimization benchmarks of increasing complexity: a geometric problem, simple-to-build QI problem, and MHD-stable QI problem. The paper provides baselines for these benchmarks using classical optimization algorithms and demonstrates that a data-driven generative model trained on the dataset can produce novel, feasible configurations without querying the expensive physics oracle.

**Additional Feedback:**

**Questions**
* Is the solver used in Section 4 one that has been used in prior literature on stellarator plasma boundary optimization?
* In Algorithm 2 (Section 5), a classifier is trained to sample from a conditional distribution, but this may be unnecessary. Could one not simply fit a GMM using only the samples $z_i$ with $y\_i=1$ and sample directly from it?
* Could the authors indicate where the experimental code used in Section 5 is publicly available?

**Comments**
* While not mandatory, comparisons with other standard black-box optimization methods such as NOMAD, CMA-ES, or Bayesian optimization would strengthen the evaluation.
* The paper should compare with standard generative models in ML, such as VAE, which are widely used and automatically perform feature extraction.
* The dataset could be used not only to train a classifier to predict feasibility but also to develop fast ML models that predict quantities such as edge rotational transform or aspect ratio. These baseline results would be helpful for ML community.

**Dataset Code Accessibility:**

Yes

**Dataset Code Comments:**

The dataset is openly available under an MIT license on Hugging Face and can be easily loaded using the datasets library. A valid Croissant metadata file is included, enabling structured access to the schema. The accompanying GitHub repository provides well-documented code for preprocessing, analysis, and benchmarking, supporting reproducibility.

**Ethical Considerations:**

No, there are no or only very minor ethics concerns

**Final Justification:**

After discussion with the authors, I have re-evaluated my initial assessment and have raised my score.

My primary concern about the lack of a comparative benchmark is now resolved. The rebuttal clarified that the paper's main objective is to provide the foundational dataset and problem definitions, not to perform an exhaustive methods comparison.

I now agree with this scope and place higher weight on the significant contribution of the dataset itself. The paper successfully provides a valuable foundation for future research, and my updated score reflects my strong support.

**Limitations Weaknesses:**

* The purpose of the baselines in Section 4 is unclear. They consist of comparisons of classical solvers used in stellarator design, but the problem is not data-driven, nor is there any indication that the dataset is being used to improve the solvers. This weakens the motivation and relevance of the baselines in the context of a data-focused benchmark.

* The data-driven pipeline in Section 5 is insufficient as a baseline. Only a single method is provided, and that method may not be appropriate.

* The work focuses exclusively on plasma boundary shape, which is only the first stage of stellarator design. It does not address the critical next step of designing the engineering coils necessary to generate the magnetic field.

**Strengths Contributions:**

* The paper provides the first large-scale public dataset focused on promising QI-type configurations, including the computed MHD equilibria.

* The authors define three well-motivated benchmark problems of varying difficulty levels, which can make a strong contribution to follow-up research.

* The dataset analysis presented in Section 2 is visually intuitive and well-presented, making it accessible even to those outside the fusion domain.

---

> ### Author Rebuttal · Authors · 2025-07-29
>
> Thank you for your thoughtful review and for recognizing ConStellaration’s pioneering dataset, well‑motivated benchmarks, and clear presentation. We’re pleased you find our contributions valuable and address each of your concerns below.
>
> ## 1. Purpose of the Classical Baselines (Section 4)
>
> Our goal with Section 4 is to anchor each benchmark using well‑established optimizers. Note that in our paper we are not prescribing necessarily a data-driven solution to our optimization benchmarks (although we believe in an opportunity for data-driven optimization). By providing strong baselines grounded in classical optimization approaches, we 1) validate that our problem formulations are non‑trivial yet solvable with trusted methods, and 2) We provide points of comparison for any future data‑driven or novel optimization approaches on exactly the same problem definitions and codebase.
>
> ## 2. Sufficiency of the Data‑Driven Pipeline (Section 5)
>
> Section 5 is intentionally a proof‑of‑concept, showcasing how the dataset enables ML methods—rather than an exhaustive survey of possible ML baselines. We expect the community to put forward approaches based on VAEs, normalizing flows, diffusion, etc to bear on these benchmarks. We consider that the depth and idiosyncrasies of applying each of these to our problems extend beyond the scope of our Dataset and Benchmark paper. We believe that our dataset (thoughtfully generated via costly optimization approaches), accessible tools for the community, and meaningful optimization benchmarks and baselines cover already significant contributions.
>
> ## 3. Focus on Plasma Boundary Only
>
> You’re correct that Stage 2—coil optimization—is essential. We deliberately scoped this paper to focus on optimizing plasma boundaries (stage one) to ensure clarity and depth for a wider audience. A comprehensive coil dataset and coil‑optimization benchmarks will be the topic of follow-up work.
>
> However, to highlight why our current `minimum magnetic gradient scale length` proxy for coil simplicity [6] effectively facilitates finding coilsets that meet reactor relevant criteria, we optimized a coilset for the solution of the “Simple-to-build QI stellarator” and obtained a coilset that meet the requirements established by reactor-relevant configurations (see the “Stellaris” power plant design [7] and table IV from [8]):
>
> | Property | Value |
> | ----- | ----- |
> | Max field error | 1.8 % |
> | Mean field error | 0.3 % |
> | Coil–plasma min distance | 1.37 m |
> | Coil–coil min distance | 1.03 m |
> | Max curvature | 1.5 m⁻¹ |
> | Total coil length | 906 m |
>
> Overall, we want to stress that the current objectives and constraints carry physics and engineering significance towards reactor-relevant configurations.
>
> ## 4. Response to Specific Questions
>
> * **Solver provenance (Section 4)**: The dominant approaches in the stellarator optimization community are gradient-based, least-squares methods where constraints are reformulated as penalty terms in the loss function. Although the Augmented Lagrangian Method (ALM) has been used as an outer-loop optimization method in similar contexts \[1\], it has been explored with such gradient-based methods and different solvers to VMEC that allow treating the equilibrium residual as a constraint. In contrast, our approach uses VMEC++ in the loop as an equilibrium solver and a derivative-free oracle (CMA-ES) within the ALM to solve the sequence of unconstrained subproblems.
>
> * **GMM trained directly on samples vs. on classifier outputs for conditional sampling (Algorithm 2\)**: Thank you for the insightful comment. While it is theoretically possible to fit a GMM directly to the latent representations of feasible samples (z\_i where y\_i \= 1\) and sample from it, we found that this approach is not sufficient in practice, because (1) feasible configurations are significantly outnumbered by infeasible ones in our dataset (or not present at all, for example, no feasible configurations for the simple-to-build QI problem are available), and fitting a GMM to this small subset leads to a model that is too broad and imprecise in the latent space. (2) The GMM trained only on feasible samples tends to place probability mass in regions far from the true feasible domain, resulting in a high rate of false positives when sampled configurations are passed to the oracle (VMEC++). To mitigate this, we incorporate an ensemble of classifiers to learn a smooth surrogate of feasibility, providing a quasi-likelihood term that reflects how confidently a point lies in the feasible region. When combined with the GMM prior in a Bayesian framework, this enables focused sampling in high-confidence feasible regions—significantly improving the precision of generated samples and reducing the number of oracle queries required to obtain valid configurations.
>
> * **Code availability (Section 5\)**: The generative‐pipeline code is already available in `notebooks/generative_model_simple_QI.ipynb` in our GitHub repo.
>
> ## 5. Feedback on Additional Comparisons and Surrogates
>
> **Other optimizers (NOMAD, CMA‑ES, Bayesian)**:
>
> * The provided optimization benchmark already includes CMA-ES (via the Nevergrad package).
> * We agree that expanding the comparison to include additional solvers would be valuable. However, our intention with this work is to provide a minimal and focused benchmark to establish a clear baseline. We hope this will serve as a foundation for future comparisons by the community.
> * Many standard derivative-free methods do not natively handle constraints. To address this, we employed an Augmented Lagrangian Method (ALM) to transform the constrained problem into a sequence of unconstrained subproblems, which are then solved using NGOpt from the Nevergrad library. In our setting, NGOpt selects CMA-ES as the underlying optimizer. This combination—ALM with CMA-ES—is a well-established strategy for handling constraints in black-box optimization [6].
>
> **Standard generative models (VAEs, flows, diffusion)**:
>
> We thank the reviewer for the suggestion. While we agree that comparing with standard generative models in machine learning—such as VAEs, normalizing flows, or diffusion models—can be valuable, the primary objective of our work is not to benchmark modeling performance. Instead, our goal is to introduce a well-defined problem statement and provide an open-access dataset tailored to this challenge, with the aim of engaging the broader ML community. To demonstrate the potential of data-driven approaches, we include a representative example rather than an exhaustive model comparison. We hope the community will explore such directions using the tools we provide.
>
> **Fast ML surrogates for physical metrics**: We agree that the dataset lends itself to train surrogate models of the metrics. Using Bayesian optimization to sweep over hyperparameters, we converged to an ensemble model of ten MLPs with three layers, 256 hidden units, and TanH activations. The MLPs mapped Fourier‐boundary coefficients to target key metrics and were trained on a 80/20% split of the dataset filtered for stellarators with three field periods. We found the following test-set results:
>
> | Metric | RMSE | R² | NRMSE | SNR |
> | ----- | ----- | ----- | ----- | ----- |
> | aspect\_ratio | 0.090 | 0.997 | 0.050 | 808.54 |
> | aspect\_ratio\_over\_edge\_rotational\_transform | 0.581 | 0.993 | 0.080 | 507.73 |
> | max\_elongation | 0.161 | 0.994 | 0.072 | 731.78 |
> | axis\_rotational\_transform\_over\_n\_field\_periods | 0.006 | 0.994 | 0.071 | 556.74 |
> | edge\_rotational\_transform\_over\_n\_field\_periods | 0.006 | 0.997 | 0.052 | 985.74 |
> | axis\_magnetic\_mirror\_ratio | 0.010 | 0.974 | 0.159 | 79.72 |
> | edge\_magnetic\_mirror\_ratio | 0.013 | 0.989 | 0.102 | 244.02 |
> | average\_triangularity | 0.018 | 0.995 | 0.065 | 806.58 |
> | vacuum\_well | 0.006 | 0.998 | 0.062 | 1934.06 |
> | minimum\_normalized\_magnetic\_gradient\_scale\_length | 0.330 | 0.990 | 0.101 | 342.57 |
> | flux\_compression\_in\_regions\_of\_bad\_curvature | 0.034 | 0.990 | 0.059 | 432.36 |
> | log\_10\_qi | 0.051 | 0.982 | 0.134 | 132.22 |
>
> While these seemingly accurate results demonstrate an efficient, accurate model of the metrics, we highlight that surrogate models are prone to extrapolation errors, particularly when queried far from the data distribution they were trained on [2,3]. Uncertainty calibration, active learning, and physics informed strategies (among others) should be considered moving forward for effective surrogate-based optimizations [4,5]. We believe that the (scientific) machine learning community can further contribute by carefully studying any of these approaches with the required depth in subsequent work.
>
> ## References
>
> [1] Rory Conlin, Patrick Kim, Daniel W Dudt, Dario Panici, and Egemen Kolemen. Stellarator optimization with constraints. Journal of Plasma Physics, 90(5):905900501, 2024\.
> [2] Shirobokov, Sergey, et al. "Black-box optimization with local generative surrogates." *Advances in neural information processing systems* 33 (2020): 14650-14662.
> [3] Duraisamy, Karthik. "Active Inference AI Systems for Scientific Discovery." *arXiv preprint arXiv:2506.21329* (2025).
> [4] Cuomo, Salvatore, et al. "Scientific machine learning through physics–informed neural networks: Where we are and what’s next." *Journal of Scientific Computing* 92.3 (2022): 88\.
> [5] Baker, Nathan, et al. *Workshop report on basic research needs for scientific machine learning: Core technologies for artificial intelligence*. USDOE Office of Science (SC), Washington, DC (United States), 2019
> [6] Paul Dufossé, Nikolaus Hansen. Augmented Lagrangian, penalty techniques and surrogate modeling for constrained optimization with CMA-ES.

---

### Official Review · Reviewer_kb3E · 2025-06-30

**Rating:** 4
**Confidence:** 5

**Summary:**

This paper introduces ConStellaration, a project designed to address a bottleneck in fusion energy research where progress in stellarator design is hindered by a lack of standardized optimization problems, baselines, and datasets. The primary contribution is a new public dataset of approximately 158,000 diverse quasi-isodynamic (QI) stellarator plasma boundaries, complete with their ideal magnetohydrodynamic (MHD) equilibria and performance metrics. To accompany the data, the authors propose three optimization benchmarks of increasing complexity: a single-objective geometric problem, a "simple-to-build" QI stellarator, and a multi-objective MHD-stable QI stellarator. For each benchmark, the project provides reference code, evaluation scripts, and strong baselines derived from classical optimization methods. Furthermore, the paper demonstrates a data-driven generative model that can produce new, feasible plasma configurations without querying an expensive physics oracle , with the overall goal of lowering the barrier for machine learning researchers to contribute to stellarator design and accelerate the path to fusion energy on the grid.

**Additional Feedback:**

How much more expensive if we consider real-world fluid flow?

**Dataset Code Accessibility:**

Yes

**Dataset Code Comments:**

The authors have taken comprehensive and transparent steps to ensure their dataset, code, and benchmarks are accessible, well-documented, and reproducible.

**Ethical Considerations:**

No, there are no or only very minor ethics concerns

**Final Justification:**

I support this dataset publication.

**Limitations Weaknesses:**

The core of the dataset and benchmarks relies on an ideal-Magnetohydrodynamics (MHD) model in a vacuum. This model treats the plasma as a perfectly conducting single fluid and does not account for critical real-world effects like finite plasma pressure, resistivity, kinetic effects, or turbulence.

**Strengths Contributions:**

This is an exceptionally strong and well-executed submission that presents a significant contribution to both the machine learning and fusion energy communities. The work is highly relevant, novel, and poised to have a substantial impact by bridging the gap between a critical, complex scientific domain and the data-driven optimization expertise of the community. You need expert to generate these data, and they are.

---

> ### Author Rebuttal · Authors · 2025-07-29
>
> Thank you for your encouraging assessment of ConStellaration. We’re happy that you recognize our work as strong, novel, well-executed, and significant for fusion and ML communities. Below we address your primary concern regarding the use of ideal‑MHD in vacuum, and then respond to your question about the cost of full, higher fidelity modelling.
>
> ## 1. Reliance on Ideal‑MHD Equilibrium in Vacuum
>
> You correctly note that our dataset consists of ideal‑MHD equilibria computed in vacuum, omitting plasma pressure physics effects. For clarity, finite‑β effects refer, at the ideal‑MHD model level, to the inclusion of non‑zero pressure profiles (assuming a null integrated toroidal current which should be negligible in precise-QI configurations [9]).
>
> Among all stellarator classes, QI configurations are inherently less sensitive to finite‑β effects [1,7], so their vacuum equilibria already serve as good proxies for modest plasma pressure cases. However, we acknowledge that including finite-beta equilibria and metrics could be valuable for the community as it 1) allows assessing the metric sensitivity to relevant plasma pressures, and 2) provides data for surrogate models spanning vacuum to finite-beta regimes.
>
> **We thus have extended our dataset to include the computed metrics and equilibria at a volume averaged β = 1%,  2%,  3%,  4%, and 5 %**. Following the Program Chair’s instructions, this extension will be available after the rebuttal period in our HuggingFace repository.
>
> ## 2. Costs of higher fidelity models
>
> Moving from ideal‑MHD vacuum solves to capture resistivity and viscous effects involve using solvers  (e.g., HINT [2], JOREK [3], SPEC [6]) that are not robust, cheap, or fast enough to generate tens of thousands of samples. While VMEC++ (ideal‑MHD, vacuum) typically requires on the order of **minutes per equilibrium** on a modern CPU core—making 158 k samples viable via a parallel HPC–, HINT or JOREK may require tens to hundreds of CPU-core hours.
>
> Other codes simulating gyrokinetic effects that provide high fidelity insights about turbulent transport like GX [4] or GENE(3D) [5] require ~O(100) GPU hours to characterize an equilibrium along a single field line at a given radial location. O(10) of such simulations are then needed to characterize the full equilibrium, leading to ~O(1000) GPU hours per equilibrium.
>
> We believe that ConStellaration strikes the right balance between fidelity, accessibility, and difficulty to make meaningful progress towards identifying reactor-relevant plasma configurations that can be later verified by any of these higher fidelity models.
>
> ## References
>
> [1] Kappel, John, Matt Landreman, and Dhairya Malhotra. "The magnetic gradient scale length explains why certain plasmas require close external magnetic coils." *Plasma Physics and Controlled Fusion* 66.2 (2024): 025018\.
> [2] Harafuji, Kenji, Takaya Hayashi, and Tetsuya Sato. "Computational study of three-dimensional magnetohydrodynamic equilibria in toroidal helical systems." Journal of computational physics 81.1 (1989): 169-192.
> [3] Hoelzl, Matthias, et al. "The JOREK non-linear extended MHD code and applications to large-scale instabilities and their control in magnetically confined fusion plasmas." *Nuclear Fusion* 61.6 (2021): 065001\.
> [4] Mandell, Noah R., et al. "GX: a GPU-native gyrokinetic turbulence code for tokamak and stellarator design." Journal of Plasma Physics 90.4 (2024): 905900402\.
> [5] Maurer, Maurice, et al. "GENE-3D: a global gyrokinetic turbulence code for stellarators." *Journal of Computational Physics* 420 (2020): 109694\.
> [6] Hudson, S. R., et al. "Computation of multi-region relaxed magnetohydrodynamic equilibria." Physics of Plasmas 19.11 (2012).
> [7] Helander, Per, and J. Nührenberg. "Bootstrap current and neoclassical transport in quasi-isodynamic stellarators." Plasma Physics and Controlled Fusion 51.5 (2009): 055004\.

---

### Official Review · Reviewer_4hs1 · 2025-07-01

**Rating:** 5
**Confidence:** 3

**Summary:**

The authors introduce ConStellaration, the first QI-like stellarator 3D plasma boundary surfaces dataset for stellarator optimization. Stellarator design is traditionally cast as a two‐stage pipeline, the first is the three‐dimensional plasma‐equilibrium magnetic field is optimized to satisfy confinement criteria. The second, electromagnetic coils are engineered to reproduce the target field. In this work, the authors focus exclusively on the first stage, in which a 3D surface parameterization defining the plasma boundary is optimized. The authors formalize three distinct optimization problems (Geometrical, Simple-to-build, MHD-stable) and try to suggest solutions using classical optimization method(trust-region constrained optimization, COBYQA, and ALM-NGOpt). Using the proposed dataset, the authors investigate the single-objective and multi-objective optimization performance by evaluating minimum normalized magnetic gradient scale length \hat{\mathcal{L}}_{\grad B} and max elongation \epsilon_{max}. The experimental results show that only ALM-NGOpt successfully identifies feasible solutions.

**Additional Feedback:**

1. Introducing explicit upper and lower limits on key evaluation metrics(minimum normalized magnetic gradient scale length \hat{\mathcal{L}}{\grad B} and max elongation \epsilon{max}) will likely improve the interpretability of the results.

2. The current optimization workflow is  running on CPU and it takes quite long execution time. Incorporating GPU‐based acceleration could markedly reduce runtime and improve the efficiency of the boundary‐surface search.

**Dataset Code Accessibility:**

Yes

**Dataset Code Comments:**

I can download the dataset from Hugging Face and access the optimization code on GitHub.

- Huggingface link: https://huggingface.co/datasets/proxima-fusion/constellaration
- Github link: https://github.com/proximafusion/constellaration

**Ethical Considerations:**

No, there are no or only very minor ethics concerns

**Final Justification:**

My questions have been addressed in the authors’ rebuttal.
Since there are no concerns, I will maintain my current score.

**Limitations Weaknesses:**

1. Since the paper focuses only on the first of the two stages, it is difficult to determine whether the optimized solutions are physically feasible and free of gaps. (I’m curious because I’m not familiar with this field.)

**Strengths Contributions:**

1. The paper has high writing quality which makes it easy to understand its motivation and background. Figures enhance the theoretical understanding of this field.

2. As far as I know, the paper is the first to define and release a dataset for stellarator optimization, which makes it easier for ML/AI researchers to engage with this domain. The defined optimization problems are intuitive.

3. The experimental framework is robustly designed, enforcing aspect‐ratio constraints A and systematically evaluating both single‐objective and multi‐objective optimization formulations.

---

> ### Author Rebuttal · Authors · 2025-07-29
>
> Thank you for your constructive feedback. We’re glad you found our writing clear, our dataset pioneering, and our experimental framework both intuitive and robust. Below we address each of your points in turn.
>
> ## 1. Physical Feasibility of Optimized Solutions
>
> Note that the configurations in our dataset are **not** intended to consist exclusively of reactor‐ready configurations. Rather, our goal is to release a **diverse ensemble of QI‑like equilibria** that can serve as a flexible playground for optimization research toward reactor‑relevant designs.
>
> The problem formulations for the “Simple-to-build QI” and “MHD-stable QI” stellarators do intend to capture aspects that enable a reactor-feasible design. To highlight how the current `minimum magnetic gradient scale length` proxy for coil simplicity [1] effectively facilitates finding coilsets that meet reactor-relevant criteria, we optimized a coilset for the solution of the “Simple-to-build QI stellarator” and obtained a coilset that meets the requirements established by reactor-relevant configurations (see the “Stellaris” power plant design [2] and table IV from [3]):
>
> | Property                    | Value   |
> |-----------------------------|---------|
> | Max field error             | 1.8%    |
> | Mean field error            | 0.3%    |
> | Coils to plasma distance    | 1.37 m  |
> | Coils to coils distance     | 1.03 m  |
> | Maximum curvature           | 1.5 m⁻¹ |
> | Total coils length          | 906 m   |
>
> Overall, we want to stress that the current objectives and constraints carry physics and engineering significance towards reactor-relevant configurations.
>
> ## 2. Explicit Bounds on Key Metrics
>
> Thank you for suggesting explicit upper and lower limits to improve interpretability. While not explicit in the paper, in code we had actually followed that suggestion: Each objective in Eq. 2 is linearly mapped onto the interval [0,  1] (higher is better). The specific bounds we use (which are available in our code repository) are:
>
> * **Max elongation** εₘₐₓ (lower is better) → mapped from [1.0,  10.0]
>
> * **Min normalized magnetic gradient scale length** (higher is better) → mapped from [0,  20.0]
>
> ## 3. CPU‑Only Workflow and GPU Acceleration
>
> Our current implementation relies on VMEC++ (as equilibrium solver) in our forward model. VMEC is inherently CPU‑based and not end‑to‑end differentiable. VMEC++ is optimized for a (multithreaded) CPU backend and no GPU backend exists today. We agree that future speed‑ups could be realized by (1) porting equilibrium solvers to GPU or (2) adopting surrogate differentiable models. We view this as a promising direction for follow‑up work once stable, GPU-compatible physics kernels become available.
>
> ______________________________________
>
> Thank you again for your positive evaluation and for these valuable suggestions. We hope our clarifications reassure you that ConStellaration establishes a strong stage 1 benchmark that translates into physically realizable, reactor‑grade designs.
>
> ## References
>
> [1] Kappel, John, Matt Landreman, and Dhairya Malhotra. "The magnetic gradient scale length explains why certain plasmas require close external magnetic coils." *Plasma Physics and Controlled Fusion* 66.2 (2024): 025018.
>
> [2] Lion, J., et al. "Stellaris: A high-field quasi-isodynamic stellarator for a prototypical fusion power plant." *Fusion Engineering and Design* 214 (2025): 114868.
>
> [3] Gil, Pedro F., Alan A. Kaptanoglu, and Eve V. Stenson. "Augmented Lagrangian methods produce cutting-edge magnetic coils for stellarator fusion reactors." *arXiv preprint arXiv:2507.12681* (2025).

---

> > ### Comment · Reviewer_4hs1 · 2025-08-05
> > **Response to author rebuttal**
> >
> > I thank the authors for their detailed response. I have read it carefully, along with the other reviews.
> > My questions have been answered. Thank you for creating such a great dataset.
> > Since there are no concerns, I will maintain my current score.

---

### Official Review · Reviewer_oDU9 · 2025-07-03

**Rating:** 5
**Confidence:** 1

**Summary:**

The "ConStellaration" project addresses the bottlenecks in stellarator design by releasing a comprehensive open dataset and a suite of optimization benchmarks. The dataset comprises approximately 158,000 diverse quasi-isodynamic (QI)-like stellarator plasma boundary shapes, each coupled with its ideal magnetohydrodynamic (MHD) equilibrium data (calculated in vacuum using VMEC++) and associated performance metrics. QI stellarators are particularly promising due to their inherent resilience to disruptions and reduced neoclassical transport.

To facilitate research, the project provides three distinct optimization benchmarks of increasing complexity:

Geometric Optimization: A single-objective problem focused on optimizing the plasma boundary's geometric properties.

"Simple-to-Build" QI Stellarator: A single-objective problem aiming to find QI configurations that are easier to construct with simpler coils.

Multi-objective Ideal-MHD Stable QI Stellarator: A more advanced, multi-objective problem that explores trade-offs between compactness and coil simplicity while ensuring MHD stability.

For each benchmark, the authors provide reference code, evaluation scripts, and strong baselines derived from classical optimization techniques. They also demonstrate how machine learning models trained on this dataset can efficiently generate new, feasible stellarator configurations without the need for expensive physics simulations. By openly releasing these resources, "ConStellaration" aims to lower the barrier to entry for optimization and machine learning researchers, accelerating progress in stellarator design and the broader goal of bringing fusion energy to the grid.

**Dataset Code Accessibility:**

Yes

**Ethical Considerations:**

No, there are no or only very minor ethics concerns

**Final Justification:**

The authors' rebuttal looks convincing.

**Limitations Weaknesses:**

Reliance on Ideal-MHD Equilibrium (in vacuum): The dataset uses ideal-MHD equilibria computed in vacuum. While a standard approach, it might not fully capture all the complex plasma physics phenomena (e.g., kinetic effects, finite beta effects, and self-consistent plasma response) that influence real stellarator performance.

Limited Scope of Optimization Objectives: While the benchmarks cover important aspects like geometry, coil simplicity, and MHD stability, stellarator design involves a multitude of other optimization objectives (e.g., turbulent transport, fast particle confinement, engineering constraints related to coil manufacturing, divertor design). The current benchmarks may not encompass the full complexity of a reactor-relevant design.

"QI-like" vs. Exact QI: The dataset focuses on "QI-like" stellarator configurations. While this provides a practical approach given the difficulty of exactly QI fields, it implies that the configurations may not be perfectly quasi-isodynamic, potentially leading to small deviations in expected performance.

Computational Cost of Data Generation: Although the dataset aims to reduce the cost of future design, the initial generation of 158,000 configurations likely involved substantial computational resources, which could be a barrier for smaller research groups to replicate or significantly expand the dataset with new physics models.

**Strengths Contributions:**

Comprehensive and Diverse Dataset: The dataset of 158,000 QI-like stellarator plasma boundaries offers a rich and varied collection, enabling robust training and testing of machine learning models.

Standardized Benchmarks: The introduction of three clearly defined optimization benchmarks, along with reference code and baselines, provides a standardized framework for comparing and evaluating different optimization algorithms and machine learning approaches. This addresses a critical need in the field.

Focus on QI Stellarators: By concentrating on QI-like stellarators, the dataset and benchmarks are directly relevant to a highly promising avenue for commercial fusion power, as QI configurations offer inherent advantages in plasma confinement and stability.

Accelerates Research through Open Science: The open-source nature of the dataset and benchmarks significantly lowers the entry barrier for new researchers, fostering interdisciplinary collaboration between fusion scientists, optimization experts, and machine learning specialists.

Potential for Surrogate Models: The demonstration that learned models can efficiently generate new, feasible configurations without expensive physics simulations highlights a significant potential for speeding up the design process.

---

> ### Author Rebuttal · Authors · 2025-07-29
>
> Thank you for your positive feedback on ConStellaration. We’re happy that you find our dataset’s breadth, the clarity of our benchmarks, and the open‐science attention to be major strengths. Below we address each of your points and clarify the intended scope of our work.
>
> ## 1. Reliance on Ideal‑MHD Equilibrium in Vacuum
>
> You correctly note that ideal‑MHD equilibria in vacuum omit plasma pressure physics effects. First, we’d like to clarify that “Kinetic effects,” “finite‑β effects,” and “self‑consistent plasma response” all refer, at the ideal‑MHD model level, to the inclusion of non‑zero pressure profiles (assuming a null integrated toroidal current which should be negligible in precise-QI configurations [9]). The plasma beta is proportional to the thermal‐to‐magnetic pressure ratio, so any positive beta value indicates a non-zero plasma pressure.
>
> Among all stellarator classes, QI configurations are inherently less sensitive to finite‑β effects [1,9], so their vacuum equilibria already serve as good proxies for modest plasma pressure cases. However, we acknowledge that including finite-beta equilibria and metrics could be valuable for the community as it 1) allows assessing the metric sensitivity to relevant plasma pressures, and 2) provides data for surrogate models spanning vacuum to finite-beta regimes.
>
> **We thus have extended our dataset to include the computed metrics and equilibria at a volume averaged β = 1%,  2%,  3%,  4%, and 5 %**. Following the Program Chair’s instructions, this extension will be available after the rebuttal period in our HuggingFace repository.
>
> While some MHD codes (e.g., HINT [2], JOREK [3], SPEC [10]) can relax ideal-MHD assumptions (finite resistivity, viscous effects), no solver today is robust, cheap, and fast enough to generate tens of thousands of samples.
>
> ## 2. Limited Scope of Optimization Objectives
>
> While our optimization objectives can’t capture the full complexity associated to the multiple systems of a reactor-relevant design, we want to clarify that several of our current metrics are effective proxies to physics phenomena raised by the reviewer:
>
> **Turbulent transport**: The “flux‑compression in regions of bad curvature” metric in our problem definitions is the (currently known) best geometrical predictor of ion temperature gradient turbulence [4] and should thus be minimized.
>
> **Fast‑particle confinement**: By targeting the QI property, we inherently suppress neoclassical losses, directly improving fast‑particle confinement [5].
>
> The natural next step is to integrate a corresponding coil dataset and coil optimization benchmarks. However,  this requires careful attention and formulation that go beyond the scope of the current work and can be the focus of a follow‑up publication. However, to highlight why our current minimum magnetic gradient scale length proxy for coil simplicity [6] effectively facilitates finding coilsets that meet reactor relevant criteria, we optimized a coilset for the solution of the “Simple-to-build QI stellarator” and obtained a coilset that meet the requirements established by reactor-relevant configurations (See the “Stellaris” power plant design [7] and table IV from [8]):
>
> | Property                    | Value   |
> |-----------------------------|---------|
> | Max field error             | 1.8%    |
> | Mean field error            | 0.3%    |
> | Coils to plasma distance    | 1.37 m  |
> | Coils to coils distance     | 1.03 m  |
> | Maximum curvature           | 1.5 m⁻¹ |
> | Total coils length          | 906 m   |
>
> Overall, we want to stress that the current objectives and constraints carry physics and engineering significance towards reactor-relevant configurations.
>
>
> ## 3. “QI‑like” vs. Exact QI
>
>
> It is true that most of our dataset configurations are not precisely QI, however, precise quasi‑isodynamic fields (i.e., magnetic fields whose deviations from an exact QI field are smaller than external contributions, e.g., the earth magnetic field) are sparse in the design space and hard to optimize for while simultaneously meeting other reactor-relevant constraints. Approximate QI already captures sufficient symmetry for excellent fast particle confinement [7, 11], and suppressed neoclassical transport as experimentally shown by Wendelstein 7-X [9].
>
> While a small fraction of our dataset configurations has very low QI residual (e.g., qi < 10^-3), In Section 5 we demonstrate how ML models trained on our data can be used to produce new equilibria that satisfy the required QI constraints of our problem definition.
>
> ## 4. Computational Cost & Reproducibility
>
> **Initial HPC investment**: Generating 158 k vacuum equilibria targeting key properties required substantial compute, but is itself a key contribution.
>
> **Full reproducibility**: We release all code and scripts; each sample can be regenerated on a standard laptop in under 1-2 hours maximum, enabling even resource‑constrained groups to extend or verify our dataset generation approaches.
>
> ___________________________________________________________________________________
>
> We hope these clarifications demonstrate that ConStellaration, being the first of its kind, strikes the right balance between breadth, fidelity, and accessibility, and we look forward to further advancing coil integration and higher‑fidelity physics in subsequent work.
>
> ## References
>
> [1] Goodman, Alan G., et al. "Constructing precisely quasi-isodynamic magnetic fields." Journal of Plasma Physics 89.5 (2023): 905890504.
>
> [2] Harafuji, Kenji, Takaya Hayashi, and Tetsuya Sato. "Computational study of three-dimensional magnetohydrodynamic equilibria in toroidal helical systems." Journal of computational physics 81.1 (1989): 169-192.
>
> [3] Hoelzl, Matthias, et al. "The JOREK non-linear extended MHD code and applications to large-scale instabilities and their control in magnetically confined fusion plasmas." Nuclear Fusion 61.6 (2021): 065001.
>
> [4] Landreman, Matt, et al. "How does ion temperature gradient turbulence depend on magnetic geometry? Insights from data and machine learning." arXiv preprint arXiv:2502.11657 (2025).
>
> [5] Goodman, Alan G., et al. "Constructing precisely quasi-isodynamic magnetic fields." Journal of Plasma Physics 89.5 (2023): 905890504.
>
> [6] Kappel, John, Matt Landreman, and Dhairya Malhotra. "The magnetic gradient scale length explains why certain plasmas require close external magnetic coils." Plasma Physics and Controlled Fusion 66.2 (2024): 025018.
>
> [7] Lion, J., et al. "Stellaris: A high-field quasi-isodynamic stellarator for a prototypical fusion power plant." Fusion Engineering and Design 214 (2025): 114868.
>
> [8] Gil, Pedro F., Alan A. Kaptanoglu, and Eve V. Stenson. "Augmented Lagrangian methods produce cutting-edge magnetic coils for stellarator fusion reactors." arXiv preprint arXiv:2507.12681 (2025).
>
> [9] Helander, Per, and J. Nührenberg. "Bootstrap current and neoclassical transport in quasi-isodynamic stellarators." Plasma Physics and Controlled Fusion 51.5 (2009): 055004.
>
> [10] Hudson, S. R., et al. "Computation of multi-region relaxed magnetohydrodynamic equilibria." Physics of Plasmas 19.11 (2012).
>
> [11] Carbajal, L., et al. "Alpha-particle confinement in Infinity Two Fusion Pilot Plant baseline plasma design." Journal of Plasma Physics (2025): 1-30.

---

### Note · Authors · 2025-08-14

We thank the reviewers for the valuable comments. We further thank the two reviewers who read and replied to our response.

We addressed reviewer kb3E concern about our data containing ONLY ideal-MHD equilibrium in vacuum by expanding the dataset to include multiple levels of plasma pressure. While we did not get a reply, we are encouraged by their original comment:
"This is an exceptionally strong and well-executed submission that presents a significant contribution to both the machine learning and fusion energy communities. The work is highly relevant, novel, and poised to have a substantial impact by bridging the gap between a critical, complex scientific domain and the data-driven optimization expertise of the community. You need expert to generate these data, and they are".

---

### Decision · Program_Chairs · 2025-09-18

**Decision:**

Accept (poster)

**Comment:**

This paper presents a dataset for supporting the development of design methods (optimization methods) of stellarators, which are magnetic confinement devices that are being pursued to deliver steady-state carbon-free fusion energy. Their design involves a high-dimensional, constrained optimization problem that requires expensive physics simulations and significant domain expertise. However, there are no standardized optimization problems with strong baselines, which is the current bottleneck for the development of this field. This paper releases an open dataset of diverse quasi-isodynamic (QI) stellarator plasma boundary shapes, paired with their ideal magnetohydrodynamic (MHD) equilibria and performance metrics. Three optimization benchmarks of varying complexity are proposed in this paper: a single-objective constrained geometric problem; a “simple-to-build” single-objective QI stellarator; and a multi-objective QI stellarator that is also MHD stable.

This paper is a well-written paper with a number of strengths such as high novelty of the dataset, high practical usefulness of the dataset, and significant contributions of this paper to the field of stellarator design (including optimization and machine learning). All reviewers support the acceptance of this paper.

Whereas some issues have been pointed out by the reviewers such as the use of an ideal-Magnetohydrodynamics (MHD) model, a limited number of objectives in the proposed optimization problems, and the high computation load of simulations, the handling of each issue has been clearly explained in the rebuttal by the authors in a convincing manner.

The average score of this paper is 4.75. Whereas one reviewer chooses "4: Borderline accept", this reviewer wrote the following one sentence in the Final Justification field: "I support this dataset publication". Thus, I think that all the four reviewers clearly support the acceptance of this paper. Since this is a good paper with respect to the presentation quality of the paper, the novelty of the dataset, and its practical usefulness (and also since all the four reviewers recommended me to accept the paper), I would like to strongly recommend the acceptance of this paper.